# Learning to live with Dale's principle: ANNs with separate excitatory and inhibitory units

**Jonathan Cornford**[1,2], **Damjan Kalajdzievski**[2,3], **Marco Leite**[4], **Amélie Lamarquette**[5]
**Dimitri M. Kullmann**[4], **Blake Richards**[1,2,†]
[1]McGill University, [2]MILA, [3]Université de Montréal, [4]UCL, [5]University of Cambridge

## Abstract

The units in artificial neural networks (ANNs) can be thought of as abstractions of biological neurons, and ANNs are increasingly used in neuroscience research. However, there are many important differences between ANN units and real neurons. One of the most notable is the absence of Dale's principle, which ensures that biological neurons are either exclusively excitatory or inhibitory. Dale's principle is typically left out of ANNs because its inclusion impairs learning. This is problematic, because one of the great advantages of ANNs for neuroscience research is their ability to learn complicated, realistic tasks. Here, by taking inspiration from feedforward inhibitory interneurons in the brain we show that we can develop ANNs with separate populations of excitatory and inhibitory units that learn just as well as standard ANNs. We call these networks Dale's ANNs (DANNs). We present two insights that enable DANNs to learn well: (1) DANNs are related to normalization schemes, and can be initialized such that the inhibition centres and standardizes the excitatory activity, (2) updates to inhibitory neuron parameters should be scaled using corrections based on the Fisher Information matrix. These results demonstrate how ANNs that respect Dale's principle can be built without sacrificing learning performance, which is important for future work using ANNs as models of the brain. The results may also have interesting implications for how inhibitory plasticity in the real brain operates.

## 1 Introduction

In recent years, artificial neural networks (ANNs) have been increasingly used in neuroscience research for modelling the brain at the algorithmic and computational level (Richards et al., 2019; Kietzmann et al., 2018; Yamins & DiCarlo, 2016). They have been used for exploring the structure of representations in the brain, the learning algorithms of the brain, and the behavioral patterns of humans and non-human animals (Bartunov et al., 2018; Donhauser & Baillet, 2020; Michaels et al., 2019; Schrimpf et al., 2018; Yamins et al., 2014; Kell et al., 2018). Evidence shows that the ability of ANNs to match real neural data depends critically on two factors. First, there is a consistent correlation between the ability of an ANN to learn well on a task (e.g. image recognition, audio perception, or motor control) and the extent to which its behavior and learned representations match real data (Donhauser & Baillet, 2020; Michaels et al., 2019; Schrimpf et al., 2018; Yamins et al., 2014; Kell et al., 2018). Second, the architecture of an ANN also helps to determine how well it can match real brain data, and generally, the more realistic the architecture the better the match (Schrimpf et al., 2018; Kubilius et al., 2019; Nayebi et al., 2018). Given these two factors, it is important for neuroscientific applications to use ANNs that have as realistic an architecture as possible, but which also learn well (Richards et al., 2019; Kietzmann et al., 2018; Yamins & DiCarlo, 2016).

Although there are numerous disconnects between ANNs and the architecture of biological neural circuits, one of the most notable is the lack of adherence to Dale's principle, which states that a neuron releases the same fast neurotransmitter at all of its presynaptic terminals (Eccles, 1976). Though there are some interesting exceptions (Tritsch et al., 2016), for the vast majority of neurons in

---

†Corresponding author: blake.richards@mcgill.ca

adult vertebrate brains, Dale's principle means that presynaptic neurons can only have an exclusively excitatory or inhibitory impact on their postsynaptic partners. For ANNs, this would mean that units cannot have a mixture of positive and negative output weights, and furthermore, that weights cannot change their sign after initialisation. In other words, a unit can only be excitatory or inhibitory. However, most ANNs do not incorporate Dale's principle.

Why is Dale's principle rarely incorporated into ANNs? The reason is that this architectural constraint impairs the ability to learn—a fact that is known to many researchers who have tried to train such ANNs, but one that is rarely discussed in the literature. However, when we seek to compare ANNs to real brains, or use them to explore biologically inspired learning rules (Bartunov et al., 2018; Whittington & Bogacz, 2019; Lillicrap et al., 2020), ideally we would use a biologically plausible architecture with distinct populations of excitatory and inhibitory neurons, and at the same time, *we would still be able to match the learning performance of standard ANNs without such constraints*.

Some previous computational neuroscience studies have used ANNs with separate excitatory and inhibitory units (Song et al., 2016; Ingrosso & Abbott, 2019; Miconi, 2017; Minni et al., 2019; Behnke, 2003), but these studies addressed questions other than matching the learning performance of standard ANNs, e.g. they focused on typical neuroscience tasks (Song et al., 2016), dynamic balance (Ingrosso & Abbott, 2019), biologically plausible learning algorithms (Miconi, 2017), or the learned structure of networks (Minni et al., 2019). Importantly, what these papers did not do is develop means by which networks that obey Dale's principle can match the performance of standard ANNs on machine learning benchmarks, which has become an important feature of many computational neuroscience studies using ANNs (Bartunov et al., 2018; Donhauser & Baillet, 2020; Michaels et al., 2019; Schrimpf et al., 2018; Yamins et al., 2014; Kell et al., 2018).

Here, we develop ANN models with separate excitatory and inhibitory units that are able to learn as well as standard ANNs. Specifically, we develop a novel form of ANN, which we call a "Dale's ANN" (DANN), based on feed-forward inhibition in the brain (Pouille et al., 2009). Our novel approach is different from the standard solution, which is to create ANNs with separate excitatory and inhibitory units by constraining whole columns of the weight matrix to be all positive or negative (Song et al., 2016). Throughout this manuscript, we refer to this standard approach as "ColumnEi" models. We have departed from the ColumnEI approach in our work because it has three undesirable attributes. First, constrained weight matrix columns impair learning because they limit the potential solution space (Amit et al., 1989; Parisien et al., 2008). Second, modelling excitatory and inhibitory units with the same connectivity patterns is biologically misleading, because inhibitory neurons in the brain tend to have very distinct connectivity patterns from excitatory neurons (Tremblay et al., 2016). Third, real inhibition can act in both a subtractive and a divisive manner (Atallah et al., 2012; Wilson et al., 2012; Seybold et al., 2015; Pouille et al., 2013), which may provide important functionality.

Given these considerations, in DANNs, we utilize a separate pool of inhibitory neurons with a distinct, more biologically realistic connectivity pattern, and a mixture of subtractive and divisive inhibition (Fig. 1). This loosely mimics the fast feedforward subtractive and divisive inhibition provided by fast-spiking interneurons in the cortical regions of the brain (Atallah et al., 2012; Hu et al., 2014; Lourenço et al., 2020). In order to get DANNs to learn as well as standard ANNs we also employ two key insights:

1. It is possible to view this architecture as being akin to normalisation schemes applied to the excitatory input of a layer (Ba et al., 2016; Ioffe & Szegedy, 2015; Wu & He, 2018), and we use this perspective to motivate DANN parameter initialisation.

2. It is important to scale the inhibitory parameter updates based on the Fisher information matrix, in order to balance the impact of excitatory and inhibitory parameter updates, similar in spirit to natural gradient approaches (Martens, 2014).

Altogether, our principle contribution is a novel architecture that obey's Dale's principle, and that we show can learn as well as standard ANNs on machine learning benchmark tasks. This provides the research community with a new modelling tool that will allow for more direct comparisons with real neural data than traditional ANNs allow, but which does not suffer from learning impairments. Moreover, our results have interesting implications for inhibitory plasticity, and provide a means for future research into how excitatory and inhibitory neurons in the brain interact at the algorithmic level.

## 2 BIOLOGICALLY INSPIRED NETWORKS THAT OBEY DALE'S PRINCIPLE

### 2.1 MODEL DEFINITION

Our design for DANNs takes inspiration from the physiology of feedforward inhibitory microcircuits in the neocortex and hippocampus. Based on these circuits, and an interpretation of layers in ANNs as corresponding to brain regions, we construct DANNs with the following architectural constraints:

1. Each layer of the network contains two distinct populations of units, an excitatory and an inhibitory population.

2. There are far fewer inhibitory units than excitatory units in each layer, just as there are far more excitatory neurons than inhibitory neurons ($\sim$ 5-10 times) in cortical regions of the brain (Tremblay et al., 2016; Hu et al., 2014).

3. As in real neural circuits where only the excitatory populations project between regions, here only excitatory neurons project between layers, and both the excitatory and inhibitory populations of a layer receive excitatory projections from the layer below.

4. All of the synaptic weights are strictly non-negative, and inhibition is enforced via the activation rules for the units (eq. 1).

5. The inhibitory population inhibits the excitatory population through a mixture of subtractive and divisive inhibition.

This constrained architecture is illustrated in Figure 1.

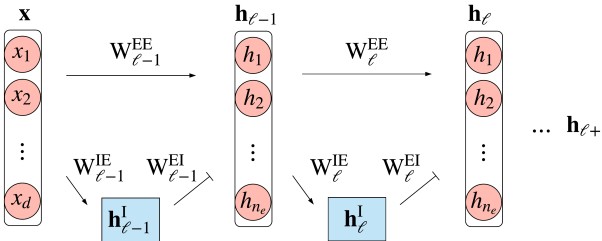

Figure 1: Illustration of DANN architecture. Lines with arrow ends indicate excitatory projections. Lines with bar ends indicate inhibitory projections, which can be *both* subtractive and divisive.

Formally, we define the network as follows. Input to the network is received as a vector of positive scalar values $\mathbf{x} \in \mathbb{R}^d_+$, which we consider to be the first excitatory population. Each hidden layer, $\ell$, is comprised of a vector of excitatory units $\mathbf{h}_\ell \in \mathbb{R}^{n_e}_+$ and inhibitory units $\mathbf{h}^I_\ell \in \mathbb{R}^{n_i}_+$, in-line with constraint (1) above. (We will drop the layer index when it is unnecessary for clarity.) Note, for the first layer ($\ell = 1$), we have $\mathbf{h}_\ell = \mathbf{x}$ and $n_e = d$. Next, based on constraint (2) we set $n_e >> n_i$, and use 10% inhibitory units as default. Following constraint (3), both the excitatory and inhibitory units receive inputs from the excitatory units in the layer below ($\mathbf{h}_{\ell-1}$), but the inhibitory units do not project between layers. Instead, excitatory units receive inputs from the inhibitory units of the same layer. In-line with constraint (4), we have three sets of strictly non-negative synaptic weights, one for the excitatory connections between layers, $\mathbf{W}^{EE}_\ell \in \mathbb{R}^{n_e \times n_e}_+$, one for the excitatory projection to the inhibitory units $\mathbf{W}^{IE}_\ell \in \mathbb{R}^{n_i \times n_e}_+$, and one for the inhibitory projections within layer $\mathbf{W}^{EI}_\ell \in \mathbb{R}^{n_e \times n_i}_+$. Finally, per constraint (5), we define the impact of the inhibitory units on the excitatory units as comprising both a subtractive and a divisive component:

$$\mathbf{h}_\ell = f(\mathbf{z}_\ell) \qquad \mathbf{z}_\ell = \frac{\mathbf{g}_\ell}{\boldsymbol{\gamma}_\ell} \odot (\mathbf{z}^E_\ell - \mathbf{W}^{EI}_\ell \mathbf{h}^I_\ell) + \boldsymbol{\beta}_\ell \tag{1}$$

$$\text{where} \quad \mathbf{z}^E_\ell = \mathbf{W}^{EE}_\ell \mathbf{h}_{\ell-1} \qquad \mathbf{h}^I_\ell = f^I(\mathbf{z}^I_\ell) = f^I(\mathbf{W}^{IE}_\ell \mathbf{h}_{\ell-1})$$

$$\boldsymbol{\gamma}_\ell = \mathbf{W}^{EI}_\ell(e^{\alpha_\ell} \odot \mathbf{h}^I_\ell)$$

where for each layer $\ell$, $\boldsymbol{\beta}_\ell \in \mathbb{R}^{n_e}$ is a bias, $\boldsymbol{g}_\ell \in \mathbb{R}^{n_e}_+$ controls the gain, $\boldsymbol{\gamma}_\ell$ is the divisive inhibitory term, and $\alpha_\ell \in \mathbb{R}^{n_i}$ is a parameter that controls the strength of this divisive inhibition. Here $\odot$ denotes elementwise multiplication (Hadamard product) and the exponential function and division are applied elementwise. In the rest of this manuscript we set $f$ to be the rectified linear function (ReLU). Though a ReLU function is not a perfect match to the input-output properties of real neurons, it captures the essential rectification operation performed by neurons in physiologically realistic low activity regimes (Salinas & Sejnowski, 2000). In this paper, we model the inhibitory units as linear (i.e. $f^I(\mathbf{z}^I) = \mathbf{z}^I$) since they receive only positive inputs and have no bias, and therefore their activation would always be in the linear part of the ReLU function. Although we make make this modelling choice mainly for mathematical simplicity, there is some biological justification, as the resting membrane potential of the class of fast-spiking interneurons most related to our model is relatively depolarised and their spike outputs can follow single inputs one-to-one (Hu et al., 2014; Galarreta & Hestrin, 2001). In future work, for example in which inhibitory connections are included between inhibitory units, we expect that the use of nonlinear functions for inhibitory units will be important.

## 3    PARAMETER INITIALISATION FOR DALE'S ANNS

In biology, excitation and inhibition are balanced (Isaacson & Scanziani, 2011), and we use this biological property to derive appropriate weight initialisation for DANNs. First we initialise excitatory parameters from an exponential distribution with rate parameter $\lambda^E$, $\mathbf{W}^{EE} \overset{iid}{\sim} \text{Exp}(\lambda^E)$, and then inhibitory parameters are initialised such that excitation and subtractive inhibition are balanced, i.e. $\mathbb{E}[z_k^E] = \mathbb{E}[(\mathbf{W}^{EI}\mathbf{z}^I)_k], \forall k$. This can be achieved in a number of ways (see appendix C.2). In line with biology, we choose to treat excitatory weights onto inhibitory and excitatory units the same, and sample $\mathbf{W}^{IE} \overset{iid}{\sim} \text{Exp}(\lambda^E)$ and set $\mathbf{W}^{EI} \leftarrow \mathbb{1}/n_i$. We note that for a DANN layer with a single inhibitory neuron, e.g. at an output layer with 10 excitatory neurons, the noise inherent in sampling a single weight vector may result in a poor match between the excitatory and inhibitory inputs, so in this case we initialise $\mathbf{W}^{IE}$ as $\frac{1}{n_e}\sum_{j=1}^{n_e}\mathbf{w}_{j,:}^{EE}$ explicitly (where $\mathbf{w}_{j,:}^{EE}$ is the $j^{th}$ row of $\mathbf{W}^{EE}$).

Next, we consider the relationship between this initialisation approach and normalisation schemes (Ba et al., 2016; Ioffe & Szegedy, 2015). Normalisation acts to both center and scale the unit activities in a network such that they have mean zero and variance one. The weight initialisation given above will produce centered activities at the start of training. We can also draw a connection between the divisive inhibition and standardisation if we assume that the elements of $\mathbf{x}$ are sampled from a rectified normal distribution, $\mathbf{x} \overset{iid}{\sim} \max(0, \mathcal{N}(0, \sigma_{\ell-1}^2))$. Under this assumption, the mean and standard deviation of the excitatory input are proportional (see Appendix D). For example, if we consider the relationship $c \cdot \mathbb{E}[z_k^E] = \text{Var}(z_k^E)^{1/2}$ for each unit $k$, we get the scalar proportionality constant $c = \sqrt{2\pi - 1}/\sqrt{d}$, as:

$$\begin{aligned}
\mathbb{E}[z_k^E] &= d \cdot \mathbb{E}[w^{EE}]\mathbb{E}[x] & \text{Var}(z_k^E) &= d \cdot \text{Var}(w^{EE})(\mathbb{E}[x^2] + \text{Var}(x)) \\
&= d \cdot \mathbb{E}[w^{EE}]\frac{\sigma_{\ell-1}}{\sqrt{2\pi}} & &= d \cdot \text{Var}(w^{EE})\sigma_{\ell-1}^2\frac{2\pi - 1}{2\pi}
\end{aligned} \tag{2}$$

with expectation over the data and the parameters, and where $w^{EE}, x$ refer to any element of $\mathbf{W}^{EE}, \mathbf{x}$. Therefore, since $\mathbb{E}[W^{EE}]^2 = \text{Var}(W^{EE})$ for weights drawn from an exponential distribution, we have

$$c = \frac{\text{Var}(z_k^E)^{\frac{1}{2}}}{\mathbb{E}[z_k^E]} = \frac{\sqrt{2\pi - 1}}{\sqrt{d}} \tag{3}$$

This proportionality means that you can perform centering and standardisation operations using the same neurons. For DANNs, $e^{\boldsymbol{\alpha}}$ will dictate the expected standard deviation of the layer's activation $\mathbf{z}$, as it controls the proportionality between subtractive and divisive inhibition for each inhibitory unit. If $e^{\boldsymbol{\alpha}}$ is set to $c$, then the divisive inhibition approximates dividing $\mathbf{z}^E$ by its standard deviation, as $\mathbb{E}[z_k^E] \cdot c = \mathbb{E}[\mathbf{w}_{k,:}^{EI}(e^{\boldsymbol{\alpha}} \odot z_k^I)] = \mathbb{E}[\gamma_k]$. We note that due to the proportionality between the mean and standard deviation of $\mathbf{z}^E$, other values of $e^{\boldsymbol{\alpha}}$ will also control the layer's variance with depth. However, given these considerations, we initialise $e^{\boldsymbol{\alpha}} \leftarrow \sqrt{2\pi - 1}/\sqrt{d}$, thereby achieving standardisation at initialisation. We find that these initialisation schemes enable DANNs to learn well. We next turn to the question of how to perform parameter updates in DANNs in order to learn well.

## 4 Parameter updates for Dale's ANNs

Unlike a layer in a column constrained network, whose affine function is restricted by sign constrained columns, a layer in a DANN is not restricted in its potential function space. This is because excitatory inputs to a layer can still have an inhibitory impact via feedforward inhibition. However, the inhibitory interneuron architecture of DANN layers introduces disparities in the degree to which updates to different parameters affect the layer's output distribution. This can be seen intuitively, for example if a single element of $\mathbf{W}^{\mathrm{IE}}$ is updated, this has an effect on each element of $\mathbf{z}$. Similarly, an update to $w_{ij}^{\mathrm{EI}}$ will change $z_i$ depending on the alignment of $x$ and all of the $j^{th}$ inhibitory unit's weights. Therefore, instead of using the euclidean metric to measure distance between parameter settings, we employ an alternative approach. Similar to natural gradient methods, we use an approximation of the Kullback-Leibler divergence (KL divergence) of the layer's output distribution for our metric. In order to help ensure that both excitatory and inhibitory parameter updates have similar impacts on the KL divergence, we scale the updates using correction terms derived below. We provide an extended derivation of these scaling factors in the Appendix E.

Given a probability distribution parameterized by some vector $\theta$, a second order approximation to the KL divergence for a change to the parameters $\theta$ is

$$\mathrm{D}_{\mathrm{KL}}\left[P(\mathbf{y}|\mathbf{x};\boldsymbol{\theta}) \,\|\, P(\mathbf{y}|\mathbf{x};\boldsymbol{\theta}+\boldsymbol{\delta})\right] \approx \frac{1}{2}\boldsymbol{\delta}^T F(\boldsymbol{\theta})\boldsymbol{\delta} \tag{4}$$

$$F(\boldsymbol{\theta}) = \underset{\mathbf{x}\sim P(\mathbf{x}),\mathbf{y}\sim P(\mathbf{y}|\mathbf{x};\boldsymbol{\theta})}{\mathbb{E}} \left[ \frac{\partial \log P(\mathbf{y}|\mathbf{x};\boldsymbol{\theta})}{\partial \boldsymbol{\theta}} \frac{\partial \log P(\mathbf{y}|\mathbf{x};\boldsymbol{\theta})}{\partial \boldsymbol{\theta}}^T \right] \tag{5}$$

Where $F(\boldsymbol{\theta})$ is the Fisher Information matrix (or just the Fisher). In order to calculate the Fisher for the parameters of a neural network, we must interpret the network's outputs in a probabilistic manner. One approach is to view a layer's activation as parameterising a conditional distribution from the natural exponential family $P(\mathbf{y}|\mathbf{x};\boldsymbol{\theta}) = P(\mathbf{y}|\mathbf{z})$, independent in each coordinate of $\mathbf{y}|\mathbf{z}$ (similar to a GLM, and as done in Ba et al. (2016)). The log likelihood of such a distribution can be written as[1]

$$\log P(\mathbf{y}|\mathbf{x};\boldsymbol{\theta}) = \frac{\mathbf{y}\cdot\mathbf{z} - \eta(\mathbf{z})}{\phi} + c(\mathbf{y},\phi) \tag{6}$$

$$\mathbb{E}[\mathbf{y}|\mathbf{x};\boldsymbol{\theta}] = f(\mathbf{z}) = \eta'(\mathbf{z}) \qquad \mathrm{Cov}(\mathbf{y}|\mathbf{x};\boldsymbol{\theta}) = \mathrm{diag}(\phi f'(\mathbf{z})) \tag{7}$$

where $f(\mathbf{z})$ is the activation function of the layer, and $\phi$, $\eta$, $c$ define the particular distribution in the exponential family. Note that here we are taking $\eta'(\mathbf{z})$ and $f'(\mathbf{z})$ to denote $\frac{\partial \eta}{\partial \mathbf{z}}$ and $\frac{\partial f}{\partial \mathbf{z}}$, respectively.

In our networks, we have used softmax activation functions at the output and ReLU activation functions in the hidden layers. In this setting, the log likelihood of the output softmax probability layer would only be defined for a one-hot vector $\mathbf{y}$ and would correspond to $\phi = 1$, $c(\mathbf{y},\phi) = 0$, and $\eta(\mathbf{z}) = \log(\sum_i e^{z_i})$. For the ReLU activation functions, the probabilistic model corresponds to a tobit regression model, in which $\mathbf{y}$ is a censored observation of a latent variable $\hat{\mathbf{y}} \sim \mathcal{N}(\mathbf{z}, \mathrm{diag}(\phi f'(\mathbf{z})))$. In this case, one could consider either the censored or pre-censored latent random variable, depending on modelling preference. As it fits well with the above framework we analyze the pre-censored random variable $\hat{\mathbf{y}}$, i.e. $f(z) = z$ in equation 6. Returning to the general case, where we consider layer's activation as parameterising a conditional distribution from the natural exponential family, the fisher of a layer is:

$$F(\boldsymbol{\theta}) = \underset{\mathbf{x}\sim P(\mathbf{x}),\mathbf{y}\sim P(\mathbf{y}|\mathbf{x};\boldsymbol{\theta})}{\mathbb{E}} \left[ \frac{\partial \mathbf{z}}{\partial \boldsymbol{\theta}} \frac{(\mathbf{y} - \eta'(\mathbf{z}))}{\phi} \frac{(\mathbf{y} - \eta'(\mathbf{z}))^T}{\phi} \frac{\partial \mathbf{z}}{\partial \boldsymbol{\theta}}^T \right] \tag{8}$$

$$= \underset{\mathbf{x}\sim P(\mathbf{x})}{\mathbb{E}} \left[ \frac{\partial \mathbf{z}}{\partial \boldsymbol{\theta}} \frac{\mathrm{diag}(f'(\mathbf{z}))}{\phi} \frac{\partial \mathbf{z}}{\partial \boldsymbol{\theta}}^T \right] \tag{9}$$

---

[1]Note the general form of the exponential family is $\log P(\mathbf{y}|\mathbf{z}) = \frac{\mathbf{z}\cdot T(\mathbf{y}) - \eta(\mathbf{z})}{\phi} + c(\mathbf{y},\phi)$, but here we only consider distributions from the natural exponential family, where $T(y) = y$, as this includes distributions of interest for us, such as Normal and Categorical, and also common distributions including Exponential, Poisson, Gamma, etc.

To estimate the approximate KL divergence resulting from the simple case of perturbing an individual parameter $\tilde{\theta} \in \boldsymbol{\theta}$ of a single-layer DANN, we only need to consider the diagonal entries of the Fisher:

$$D_{\mathrm{KL}}\big[P_{\boldsymbol{\theta}} \,\|\, P_{\boldsymbol{\theta}+\boldsymbol{\delta}_{\tilde{\theta}}}\big] \approx \frac{\delta^2}{2\phi} \sum_k^{n_e} \mathop{\mathbb{E}}_{\mathbf{x} \sim P(\mathbf{x})} \left[ f'(z_k)(\frac{\partial z_k}{\partial \tilde{\theta}})^2 \right] \tag{10}$$

where $\boldsymbol{\delta}_{\tilde{\theta}}$ represents a 1-hot vector corresponding to $\tilde{\theta}$ multiplied by a scalar $\delta$. We now consider the approximate KL divergence after updates to a single element of $\mathbf{W}^{\mathrm{EE}}$, $\mathbf{W}^{\mathrm{IE}}$, $\mathbf{W}^{\mathrm{EI}}$ and $\boldsymbol{\alpha}$:

$$D_{\mathrm{KL}}\big[P_{\boldsymbol{\theta}} \,\|\, P_{\boldsymbol{\theta}+\boldsymbol{\delta}_{\mathbf{W}_{ij}^{\mathrm{EE}}}}\big] \approx \frac{\delta^2}{2\phi} \mathbb{E}\left[ f'(z_i)(\frac{g_i}{\gamma_i}x_j)^2 \right] \tag{11}$$

$$D_{\mathrm{KL}}\big[P_{\boldsymbol{\theta}} \,\|\, P_{\boldsymbol{\theta}+\boldsymbol{\delta}_{\mathbf{W}_{ij}^{\mathrm{IE}}}}\big] \approx \frac{\delta^2}{2\phi} \sum_k^{n_e} \mathbb{E}\left[ f'(z_k)(\frac{g_k}{\gamma_k}x_j)^2 (w_{ki}^{\mathrm{EI}}a_{ki})^2 \right] \tag{12}$$

$$D_{\mathrm{KL}}\big[P_{\boldsymbol{\theta}} \,\|\, P_{\boldsymbol{\theta}+\boldsymbol{\delta}_{\mathbf{W}_{ij}^{\mathrm{EI}}}}\big] \approx \frac{\delta^2}{2\phi} \sum_n^d \mathbb{E}\left[ f'(z_i)(\frac{g_i}{\gamma_i}x_n)^2 (w_{jn}^{\mathrm{IE}}a_{ij})^2 \right] \tag{13}$$

$$+ \frac{\delta^2}{2\phi} \sum_{n \neq m}^d \mathbb{E}\left[ f'(z_i)(\frac{g_i}{\gamma_i})^2 x_n x_m w_{jn}^{\mathrm{IE}} w_{jm}^{\mathrm{IE}} (a_{ij})^2 \right]$$

$$D_{\mathrm{KL}}\big[P_{\boldsymbol{\theta}} \,\|\, P_{\boldsymbol{\theta}+\boldsymbol{\delta}_{\alpha_i}}\big] = \frac{\delta^2}{2\phi} \sum_k^{n_e} \sum_j^d \mathbb{E}\left[ f'(z_k)(\frac{g_k}{\gamma_k}x_j)^2 w_{ki}^{\mathrm{EI}} w_{ij}^{\mathrm{IE}} (a_{ki}-1)^2 \right] \tag{14}$$

$$+ \frac{\delta^2}{2\phi} \sum_k^{n_e} \sum_{n \neq m}^d \mathbb{E}\left[ f'(z_k)(\frac{g_k}{\gamma_k})^2 x_n x_m w_{in}^{\mathrm{IE}} w_{im}^{\mathrm{IE}} (w_{ki}^{\mathrm{EI}})^2 (a_{ki}-1)^2 \right]$$

Where $a_{kj} = \frac{e^{\alpha_j}}{\gamma_k}(z_k^{\mathrm{E}} - (\mathbf{W}^{\mathrm{EI}}z^{\mathrm{I}})_k) + 1$, and expectations are over the data, $\mathbb{E}_{\mathbf{x} \sim P(\mathbf{x})}$.

Therefore, as a result of the feedforward inhibitory architecture of DANNs, for a parameter update $\delta$, the effect on the model's distribution will be different depending on the updated parameter-type. While the exact effect depends on the degree of co-variance between terms, the most prevalent differences between and within the excitatory and inhibitory parameter-types are the sums over layer input and output dimensions. For example, an inhibitory weight update of $\delta$ to $w_{ij}^{\mathrm{IE}}$ is expected to change the model distribution approximately $n_e$ times more than an excitatory weight update of $\delta$ to $w_{ij}^{\mathrm{EE}}$. In order to balance the impact of updating different parameter-types, we update DANN parameters after correcting for these terms: updates to $\mathbf{W}^{\mathrm{IE}}$ were scaled by $\sqrt{n_e}^{-1}$, $\mathbf{W}^{\mathrm{EI}}$ by $d^{-1}$ and $\boldsymbol{\alpha}$ by $(d\sqrt{n_e})^{-1}$. As a result, inhibitory unit parameters updates are scaled down relative to excitatory parameter updates. This leads to an interesting connection to biology, because while inhibitory neuron plasticity is well established, the rules and mechanisms governing synaptic updates are different from excitatory cells (Kullmann & Lamsa, 2007; Kullmann et al., 2012), and historically interneuron synapses were thought to be resistant to long-term weight changes (McBain et al., 1999).

Next, we empirically verified that our heuristic correction factors captured the key differences between parameter-types in their impact on the KL divergence. To do this we compared parameter gradients before and after correction, to parameter gradients multiplied by an approximation of the diagonal of the Fisher inverse for each layer (which we refer to as Fisher corrected gradients), see Appendix F.3. The model was trained for 50 epochs on MNIST, and updated using the Fisher corrected gradients. Throughout training, we observed that the heuristic corrected gradients were more aligned to the Fisher corrected gradients than the uncorrected gradients were (Fig. 2). Thus, our derived correction factors help to balance the impact of excitatory and inhibitory updates on the network's behaviour. Below, we demonstrate that these corrections are key to getting DANNs to learn well.

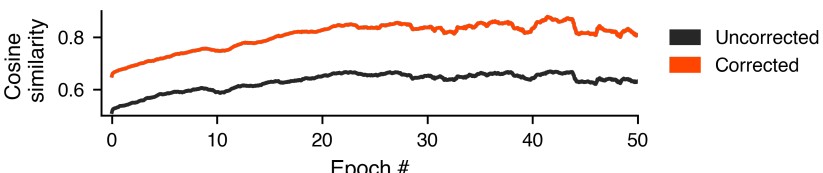

Figure 2: Empirical verification of update correction terms: Cosine of the angle between gradients multiplied by an approximation of the diagonal of the Fisher inverse for each layer, and either uncorrected gradients (black) or corrected gradients (orange) over 50 epochs. Plot displays a moving average over 500 updates

## 5 EXPERIMENTAL RESULTS

Having derived appropriate parameter initialisation and updates for DANNs, we now explore how they compare to traditional ANNs and ColumnEi models on simple benchmark datasets. In brief, we find that column constrained models perform poorly, failing even to achieve zero training-set error, whereas DANNs perform equivalently to traditional ANNs.

### 5.1 IMPLEMENTATION DETAILS

All models were composed of 4 layers: in general 3 hidden layers of dimension 500 with a ReLU activation function followed by a softmax output with 10 units, and all experiments were run for 50 epochs with batch size 32. Unless stated, for DANNs and ColumnEi models, 50 inhibitory units were included per hidden layer. For DANN models, the softmax output layer was constructed with one inhibitory unit. For ColumnEi models, each hidden layer's activation is $\mathbf{z} = \mathbf{W}\mathbf{x}$ where 500 columns of $\mathbf{W}$ were constrained to be positive and 50 negative (therefore for ColumnEi models $\mathbf{h}_\ell$ was of dimension 550). ColumnEi layer weights were initialised so that variance did not scale with depth and that activations were centered (see Appendix C.1 for further details). All benchmark datasets (MNIST, Kuzushiji MNIST and Fashion MNIST) were pre-processed so that pixel values were in $[0, 1]$. Learning rates were selected according to validation error averaged over 3 random seeds, after a random search (Orion; Bouthillier et al. (2019), log uniform [10, 1e-5], 100 trials, 10k validation split). Selected models were then trained on test data with 6 random seeds. Plots show mean training error per epoch, and mean test set error every 200 updates over random seeds. Tables show final error mean $\pm$ standard deviation. For further implementation details and a link to the accompanying code see Appendix F.

Note that because our goal in this paper is not to achieve state-of-the-art performance, we did not apply regularisation techniques, such as dropout and weight decay, or common modifications to stochastic gradient descent (SGD). Instead the goal of the experiments presented here was simply to determine whether, in the simplest test case scenario, DANNs can learn better than ColumnEi models and as well as traditional ANNs.

### 5.2 COMPARISON OF DANNS TO COLUMN-EI MODELS AND MLPS

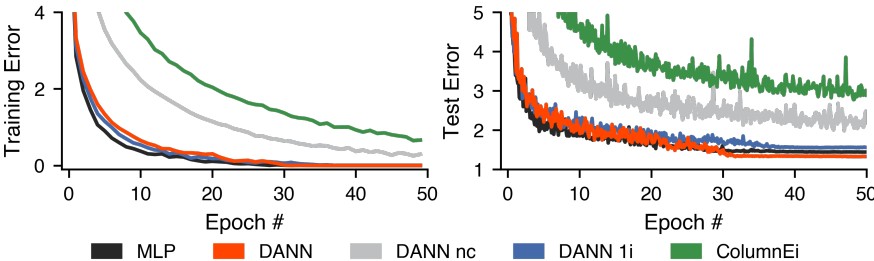

Figure 3: Model comparison on MNIST dataset. *nc* - no update corrections, *1i* - one inhibitory unit

We first compared model performance on the MNIST dataset (Fig 3). We observed that ColumnEi models generalised poorly, and failed to achieve 0 % training error within the 50 epochs. This confirms the fact that such models cannot learn as well as traditional ANNs. In contrast, we observed that DANNs performed equivalently to multi-layer perceptrons (MLPs), and even generalised marginally better. This was also the case for ColumnEi and DANN models constructed with more inhibitory units (Supp. Fig. 6, 100 inhibitory units per layer). In addition, performance was only slightly worse for DANNs with one inhibitory unit per layer. These results show that DANN performance generalizes to different ratios of excitatory-to-inhibitory units. We also found that not correcting parameter updates using the corrections derived from the Fisher significantly impaired optimization, further verifying the correction factors (Fig 3).

Next, we compared DANN performance to MLPs trained with batch and layer normalization on more challenging benchmark datasets (Fig 4). Again we found that DANNs performed equivalently to these standard architectures, whereas ColumnEi models struggled to achieve acceptable performance.

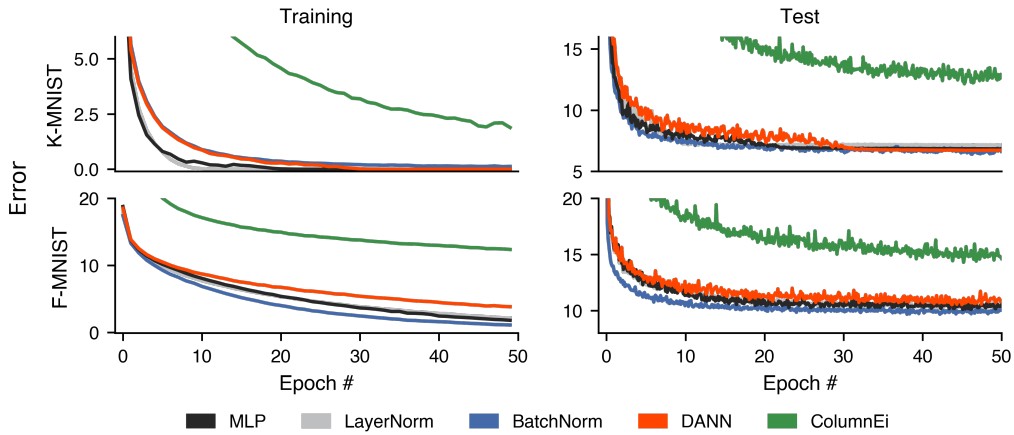

Figure 4: Model comparison on Fashion MNIST and Kuzushiji MNIST datasets.

We also explored methods for improving DANN performance (Appendix F.4). First, in order to maintain the positive DANN weight constraint, if after a parameter update a weight was negative, we reset it to zero, i.e. $\boldsymbol{\theta} \leftarrow \max(0, \boldsymbol{\theta})$, and as a result the actual update is no longer that suggested by SGD. We therefore experimented with temporarily reducing the learning rate whenever this parameter clipping would reduce the cosine of the angle made between the gradient and actual updates below a certain constraint (see Appendix F.4). Second, we note that the divisive inhibition term, $\boldsymbol{\gamma}$, appears in the denominator of the weight gradients (Appendix E.2) and, therefore, if $\boldsymbol{\gamma}$ becomes small, the gradients will become large, potentially resulting in inappropriate parameter updates. We therefore wondered if constraining the gradient norm would be particularly effective for DANNs. We tested both of these modifications to DANNs trained on Fashion MNIST (Supp. Fig. 5). However, we found that they provided no observable improvement, indicating that the loss landscape and gradients were well behaved over optimization.

Finally, we provide an analysis and preliminary experiments detailing how the DANN architecture described above may be extended to recurrent and convolutional neural networks in future work (Appendix B). In brief, we unroll recurrent networks over time and place inhibition between both network layers and timesteps, corresponding to fast feedforward and local recurrent inhibition, respectively. For convolutional architectures, we can directly apply the DANN formulation to activation maps if inhibitory and excitatory filters are of the same size and stride. Supporting this, we found that a DANN version of VGG16 (Simonyan & Zisserman, 2014) converged equivalently to a standard VGG16 architecture (Supp.Fig.7).

Altogether, our results demonstrate that: (1) the obvious approach to creating ANNs that obey Dale's principle (ColumnEi models) do not learn as well as traditional ANNs, (2) DANNs learn better than ColumnEi models and as well as traditional ANNs, (3) DANN learning is significantly improved by taking appropriate steps to scale updates in excitatory and inhibitory units appropriately.

## 6 DISCUSSION

Here we presented DANNs, a novel ANN architecture with separate inhibitory and excitatory units. We derived appropriate parameter initialisation and update rules and showed experimentally that, unlike ANNs where some columns are simply constrained to be positive or negative, DANNs perform equivalently to traditional ANNs on benchmark datasets. These results are important as they are, as far as we know, the first example of an ANN architecture that fully adheres to Dale's law *without sacrificing learning performance*. However, our results also raise an interesting question: why does nature employ Dale's principle? After all, we did not see any *improvement* over normal ANNs in our experiments. There are two possible hypotheses. First, it is possible that Dale's principle represents an evolutionary local minima, whereby early phylogenetic choices led to constraints on the system that were difficult to escape via natural selection. Alternatively, Dale's principle may provide some computational benefit that we were unable to uncover given the specific tasks and architectures we used here. For example, it has been hypothesized that inhibition may help to prevent catastrophic forgetting (Barron et al., 2017). We consider exploring these questions an important avenue for future research.

There are a number of additional avenues for future work building upon DANNs, the most obvious of which are to further extend and generalize DANNs to recurrent and convolution neural networks (see Appendix B). It would also be interesting to explore the relative roles of subtractive and divisive inhibition. While subtractive inhibition is required for the unconstrained functional space of DANN layers, divisive inhibition may confer some of the same optimisation benefits as normalisation schemes. A related issue would be to explore the continued balance of excitation and inhibition during optimization, because while DANNs are initialised such that these are balanced, and inhibition approximates normalisation schemes, the inhibitory parameters are updated during training, and the model is free to diverge from this initialisation. As a result, the distribution of layer activations may be unstable over successive parameter updates, potentially harming optimization. In the brain, a variety of homeostatic plasticity mechanisms stabilize neuronal activity. For example, reducing excitatory input naturally results in a reduction in inhibition in real neural circuits (Tien & Kerschensteiner, 2018). It would therefore be interesting to test the inclusion of a homeostatic loss to encourage inhibition to track excitation throughout training. Finally, we note that while fast feedforward inhibition in the mammalian cortex was the main source of inspiration for this work, future investigations may benefit from drawing on a broader range of neurobiology, for example by incorporating principles of invertebrate neural circuits, such as the mushroom bodies of insects (Serrano et al., 2013).

In summary, DANNs sit at the intersection of a number of programs of research. First, they are a new architecture that obeys Dale's principle, but which can still learn well, allowing researchers to more directly compare trained ANNs to real neural data (Schrimpf et al., 2018; Yamins et al., 2014). Second, DANNs contribute towards computational neuroscience and machine learning work on inhibitory interneurons in ANNs, and in general towards the role of inhibitory circuits and plasticity in neural computation (Song et al., 2016; Sacramento et al., 2018; Costa et al., 2017; Payeur et al., 2020; Atallah et al., 2012; Barron et al., 2017). Finally, the inhibition in DANNs also has an interesting connection to normalisation methods used to improving learning in deep networks (Ioffe & Szegedy, 2015; Wu & He, 2018; Ba et al., 2016). As DANNs tie these distinct programs of research together into a single model, we hope they can serve as a basis for future research at the intersection of deep learning and neuroscience.

## ACKNOWLEDGEMENTS

We would like to thank Shahab Bakhtiari, Luke Prince, and Arna Ghosh for their helpful comments on this work. This work was supported by grants to BAR, including a NSERC Discovery Grant (RGPIN-2020-05105), an Ontario Early Career Researcher Award (ER17-13-242), a Healthy Brains, Healthy Lives New Investigator Start-up (2b-NISU-8), and funding from CIFAR (Learning in Machines and Brains Program, Canada CIFAR AI Chair), the Wellcome Trust and by the Medical Research Council (UK). Additionally, DK was supported by the FRQNT Strategic Clusters Program (2020-RS4-265502-UNIQUE).

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

# Supplementary Material

## A    Supplementary results

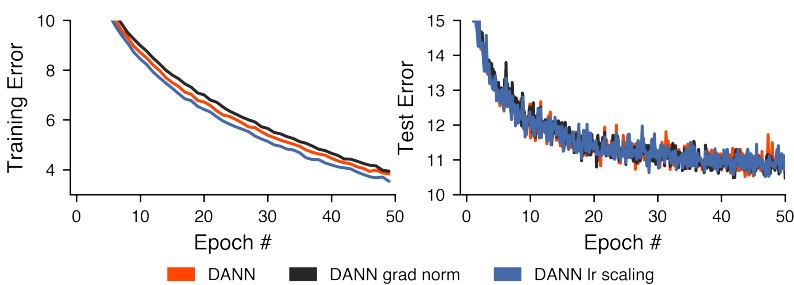

Figure 5: DANNs trained on Fashion MNIST with gradient normalisation and learning rate scaling.

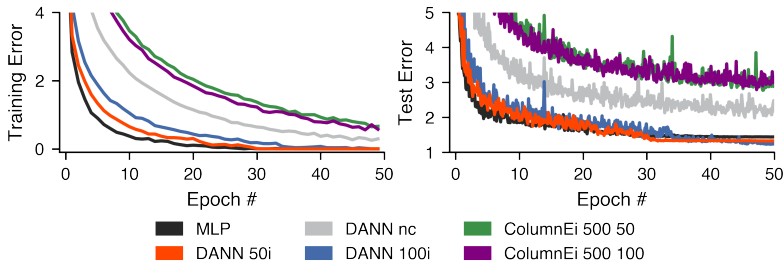

Figure 6: Model comparison on MNIST dataset as in Fig. 3 but including models with 100 inhibitory units. *nc* - no update corrections, *#i* - no. inhibitory units, ColumnEi #e #i.

Table 1: MNIST results

| Model | # inhib | Learning rate | Train error | Test error |
|---|---|---|---|---|
| MLP | 0 | 0.2976 | $0.0 \pm 0.0$ | $1.44 \pm 0.034$ |
| DANN | 1 | 0.05327 | $0.0 \pm 0.0$ | $1.56 \pm 0.041$ |
| DANN | 50 | 0.1107 | $0.0 \pm 0.0$ | $1.325 \pm 0.066$ |
| DANN | 100 | 0.3576 | $0.0 \pm 0.0$ | $1.244 \pm 0.067$ |
| DANN nc | 50 | 0.003981 | $0.293 \pm 0.039$ | $2.167 \pm 0.113$ |
| ColumnEi | 50 | 0.05273 | $0.662 \pm 0.071$ | $3.035 \pm 0.231$ |
| ColumnEi | 100 | 0.06533 | $0.574 \pm 0.092$ | $2.857 \pm 0.08$ |

Table 2: K-MNIST results

| Model | # inhib | Learning rate | Train error | Test error |
|---|---|---|---|---|
| MLP | 0 | 0.213 | $0.0 \pm 0.0$ | $6.842 \pm 0.119$ |
| LayerNorm | 0 | 0.1166 | $0.0 \pm 0.0$ | $7.171 \pm 0.134$ |
| BatchNorm | 0 | 0.7235 | $0.123 \pm 0.017$ | $6.747 \pm 0.129$ |
| DANN | 50 | 0.156 | $0.0 \pm 0.0$ | $6.728 \pm 0.23$ |
| ColumnEi | 50 | 0.05426 | $1.886 \pm 0.191$ | $12.763 \pm 0.44$ |

Table 3: Fashion MNIST results

| Model | # inhib | Learning rate | Train error | Test error |
|---|---|---|---|---|
| MLP | 0 | 0.08403 | $1.793 \pm 0.119$ | $10.626 \pm 0.466$ |
| LayerNorm | 0 | 0.04743 | $2.136 \pm 0.076$ | $10.445 \pm 0.455$ |
| BatchNorm | 0 | 0.1098 | $1.104 \pm 0.044$ | $9.992 \pm 0.218$ |
| DANN | 50 | 0.01973 | $3.832 \pm 0.031$ | $10.962 \pm 0.365$ |
| ColumnEi | 50 | 0.02265 | $12.365 \pm 0.217$ | $14.986 \pm 0.674$ |

## B  EXTENSION OF DANNs TO OTHER ARCHITECTURES

Here we discuss how our results and analysis of fully-connected feedforward Dale's ANNs may be applied to convolutional and recurrent neural networks.

### B.1  EXTENSION TO CONVOLUTIONAL NEURAL NETWORKS

Consider the response of a standard convolutional layer of $n$ output channels with filters of size $k \times k$ at a single position $j$ over $m$ input channels:

$$\mathbf{z}_j = \mathbf{W}\mathbf{x}_j + \mathbf{b} \tag{15}$$

Here, $\mathbf{W}$ is a $n \times k^2 m$ matrix whose rows correspond to the kernel weights of each output channel, and the vector $\mathbf{x}_j$ of length $k^2 m$ contains the values over the $n$ input channels for the spatial location $i$. Concatenating each input location $\mathbf{x}_j$ as the columns of a matrix $\mathbf{X}$, the full output of the convolutional layer over all input locations can be expressed as $\mathbf{Z} = \mathbf{W}\mathbf{X} + \mathbf{b}$, where $\mathbf{b}$ is broadcast over the columns of $\mathbf{Z}$. We can readily make an equivalent DANN formulation for a convolution layer by assuming the same kernel size and stride for excitatory and inhibitory filter-sets $\mathbf{W}^{\text{EE}}$ and $\mathbf{W}^{\text{EI}}$:

$$\mathbf{z}_j = \frac{\mathbf{g}}{\boldsymbol{\gamma}} \odot (\mathbf{W}^{\text{EE}}\mathbf{x}_j - \mathbf{W}^{\text{EI}}\mathbf{W}^{\text{IE}}\mathbf{x}_j) + \boldsymbol{\beta},$$
$$\boldsymbol{\gamma} = \mathbf{W}^{\text{EI}}(e^{\boldsymbol{\alpha}} \odot \mathbf{W}^{\text{IE}}\mathbf{x}_j) \tag{16}$$

Here the inhibitory channels are mapped to each excitatory output channel by $\mathbf{W}^{\text{IE}}$ for subtractive inhibition, and are first scaled by $e^{\boldsymbol{\alpha}}$ for divisive inhibition. For parameter initialisation, by following the approach of He et al. (2015) and considering the response of the layer at a single location, we use the same initialisations as those derived in section 3, but where the input dimension $d$ is the product of kernel size and input channels, $k^2 m$. Next, the correction factors to parameters updates apply as in section 4 as the KL divergence is summed over each valid input location $j$, which results in approximately the same multiplicative factor for each parameter, but does not change the approximate relative differences between parameter types:

$$\mathrm{D}_{\mathrm{KL}}\big[P_{\boldsymbol{\theta}} \,\|\, P_{\boldsymbol{\theta}+\boldsymbol{\delta}_{\tilde{\theta}}}\big] \approx \sum_j \left( \frac{\delta^2}{2\phi} \sum_i^n \mathop{\mathbb{E}}_{\mathbf{x} \sim P(\mathbf{x})} \left[ f'(z_{i,j}) \big(\frac{\partial z_{i,j}}{\partial \tilde{\theta}}\big)^2 \right] \right) \tag{17}$$

where we consider the full response $\mathbf{Z}$ of the layer over all valid kernel locations.

In order to confirm our extension to convolutional neural networks we conducted preliminary experiments with DANN versions of convolutional neural networks as described above. Below, we show results of training a standard VGG16 architecture, and a DANN version of the VGG16 architecture (Supp. Fig. 7) on CIFAR-10. As can be seen, the DANN network trains approximately as well as the standard VGG16 model.

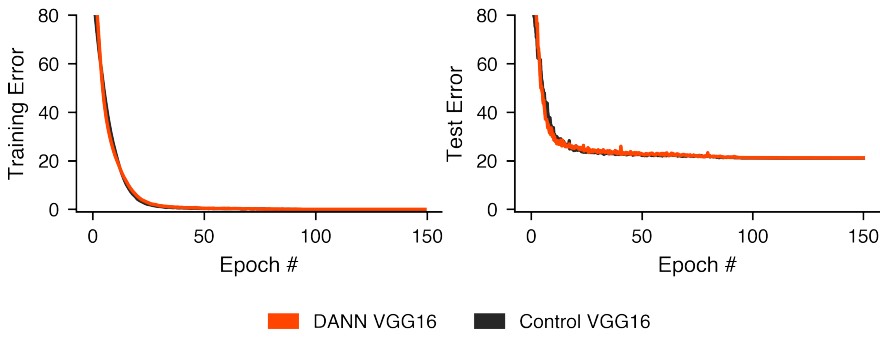

Figure 7: Convolutional network results on CIFAR-10 with control and DANN VGG16 models. Plots show mean training and test set error over 6 random seeds.

Both control and DANN VGG16 architectures were trained on CIFAR-10 with stochastic gradient descent with batch size 128, without dataset augmentation, dropout, or batch normalisation layers. Best model learning rates (control - 0.089 , DANN - 0.03458) were selected after a random search according to average final validation error over random seeds, and conditional on all seeds beginning to converge within 5 epochs (convergence defined as validation error $< 90\%$). The random search was performed with learning rates sampled from a log-uniform [1e-4,1] distribution, 3 seeds per trial, 60 trials, 150 epochs, and with a 10k validation split. Final epoch test error over 6 random seeds was $21.08 \pm 0.811$ for the control VGG16 model, and $21.178 \pm 0.348$ for the DANN-VGG16 model with constrained weights. VGG16 models were adapted from code here[2], and for the DANN VGG16 model we used 10 inhibitory filters per 64 excitatory filters, and 10% inhibitory units in the fully connected layers.

## B.2 Extension to Recurrent Neural Networks

We can readily make a connection between the fully-connected Dales ANNs described in Section 2.1 and recurrent neural networks (RNNs) by considering the similarities between depth and time. As has been previously noted, a shallow RNN unrolled over time can be expressed as a deep neural network with weight sharing (Liao & Poggio, 2016).

$$\mathbf{h}_t = f(\mathbf{z}_t) \qquad \mathbf{z}_t = \frac{\mathbf{g}_t}{\gamma_t} \odot \hat{\mathbf{W}} \mathbf{h}_{t-1} + \boldsymbol{\beta} \tag{18}$$

$$\text{where} \quad \hat{\mathbf{W}} = \mathbf{W}^{\text{EE}} - \mathbf{W}^{\text{EI}} \mathbf{W}^{\text{EI}}, \quad \gamma_t = \mathbf{W}^{\text{EI}}(e^\alpha \odot \mathbf{W}^{\text{EI}} \mathbf{h}_{t-1})$$

where in this simple case, recurrent processing steps over time are applied to the input $\mathbf{x} = \mathbf{h}_0$. In this view, layer depth corresponds to time, and inhibition between layers corresponds to fast feedback inhibition.

We note that if there are a sequence of inputs coming at each time-step, $x_t$, then this formulation can still hold, but with a simple modification to incorporate the time-varying inputs. Specifically, we need to add additional input weights, $\hat{\mathbf{U}}$:

$$\mathbf{h}_t = f(\mathbf{z}_t) \qquad \mathbf{z}_t = \frac{\mathbf{g}_t}{\gamma_t} \odot \hat{\mathbf{W}} \mathbf{h}_{t-1} + \frac{\mathbf{g_x}}{\gamma_\mathbf{x}} \odot \hat{\mathbf{U}} \mathbf{x}_t + \boldsymbol{\beta} \tag{19}$$

---

[2]https://github.com/pytorch/vision/blob/master/torchvision/models/vgg.py

$$\text{where} \quad \hat{\mathbf{W}} = \mathbf{W}^{\text{EE}} - \mathbf{W}^{\text{EI}}\mathbf{W}^{\text{EI}}, \quad \boldsymbol{\gamma}_t = \mathbf{W}^{\text{EI}}(e^{\alpha_t} \odot \mathbf{W}^{\text{EI}}\mathbf{h}_{t-1})$$
$$\hat{\mathbf{U}} = \mathbf{U}^{\text{EE}} - \mathbf{U}^{\text{EI}}\mathbf{U}^{\text{EI}}, \quad \boldsymbol{\gamma}_{\mathbf{x}} = \mathbf{U}^{\text{EI}}(e^{\alpha_{\mathbf{x}}} \odot \mathbf{U}^{\text{EI}}\mathbf{x}_t)$$

All of the existing DANN approaches developed above can be applied to this case.

## C  PARAMETER INITIALISATIONS

In this section we provide further details regarding parameter initialisations.

Throughout we assume that the elements of the input, $\mathbf{x}$, to a layer $\ell$ are $iid$ and also the output of a layer $\ell - 1$ whose pre-activations were distributed $\mathcal{N}(0, \sigma_{\ell-1}^2)$. Therefore $x$ will follow a rectified normal distribution:

$$\mathbb{E}[x] = \int_0^\infty x \cdot \frac{e^{-x^2/2\sigma_{\ell-1}^2}}{\sigma_{\ell-1}\sqrt{2\pi}}dx = \frac{\sigma_{\ell-1}}{\sqrt{2\pi}}$$
$$\mathbb{E}[x^2] = \int_0^\infty x^2 \cdot \frac{e^{-x^2/2\sigma_{\ell-1}^2}}{\sigma_{\ell-1}\sqrt{2\pi}}dx = \frac{\sigma_{\ell-1}^2}{2} \tag{20}$$
$$\text{Var}(x) = \mathbb{E}[x^2] - \mathbb{E}[x]^2 = \sigma_{\ell-1}^2\frac{\pi - 1}{2\pi}$$
$$\text{Var}(x) + \mathbb{E}[x]^2 = \sigma_{\ell-1}^2\frac{2\pi - 1}{2\pi}$$

where here, and throughout the text, non-indexed non-bold to refers to any element of a vector or matrix, e.g $\mathbb{E}[x]$ refers to the expectation of any element of $\mathbf{x}$, $\text{Var}(x)$ refers to the variance of any element of $\mathbf{x}$, etc. In addition, for all models we draw positively constrained weights $iid$ from exponential distributions, and make use of the following properties for $w \sim \text{Exp}(\lambda)$

$$\mathbb{E}[w] = \frac{1}{\lambda}$$
$$\text{Var}(w) = \frac{1}{\lambda^2} = \mathbb{E}[w]^2 \tag{21}$$
$$\mathbb{E}[w^2] = \frac{2!}{\lambda^2} = 2\text{Var}(w) = 2\mathbb{E}[w]^2$$

### C.1  COLUMN CONSTRAINED EI MODELS AND WEIGHT INITIALISATION

Here we provide detail on the parameter initialisation of column constrained models. Layer activations are $\mathbf{z} = \mathbf{W}\mathbf{x}$ where columns of $\mathbf{W}$ are constrained to be positive or negative. Therefore, for convenience, let us denote $\mathbf{W} = [\mathbf{W}^+, \mathbf{W}^-]$, and $\mathbf{x} = [\mathbf{x}^E, \mathbf{x}^I]$, and we assume $\mathbf{x}_i^E, \mathbf{x}_j^I$ are $iid$ $\forall i, j$. Note for this model, $n_e + n_i = d$, the input dimensionality. As for DANN models, throughout training we preserve the sign constraints of the weights by resetting weights using rectification around zero, i.e. $\mathbf{W}^+ \leftarrow \max(0, \mathbf{W}^+)$, $\mathbf{W}^- \leftarrow \min(0, \mathbf{W}^-)$.

At initialisation for the column constrained model for each layer we require $\mathbb{E}[z_k] = 0$, $\text{Var}(z_k) = \sigma_{\ell-1}^2$.

$$\mathbb{E}[z_k] = n_e\mathbb{E}[w^+]\mathbb{E}[x] - n_i\mathbb{E}[w^-]\mathbb{E}[x]$$
$$n_e\mathbb{E}[w^+]\mathbb{E}[x] = n_i\mathbb{E}[w^-]\mathbb{E}[x] \tag{22}$$
$$\mathbb{E}[w^-] = \mathbb{E}[w^+]\frac{n_e}{n_i}$$

Where $w^+, w^-$ refer to any element of $\mathbf{W}^+, \mathbf{W}^-$.

$$
\begin{aligned}
\operatorname{Var}(z_k) &= \sum_i^{n_e} \operatorname{Var}(w_{ki}^+ x_i^E) + \sum_j^{n_i} \operatorname{Var}(w_{kj}^- x_j^I) \\
&= n_e \operatorname{Var}(w^+ x) + n_i \operatorname{Var}(w^- x) \\
&= n_e \big( \mathbb{E}[w^+]^2 \operatorname{Var}(x) + \operatorname{Var}(w^+)\mathbb{E}[x]^2 + \operatorname{Var}(w^+)\operatorname{Var}(x) \big) \\
&\quad + n_i \big( \mathbb{E}[w^-]^2 \operatorname{Var}(x) + \operatorname{Var}(w^-)\mathbb{E}[x]^2 + \operatorname{Var}(w^-)\operatorname{Var}(x) \big)
\end{aligned}
\tag{23}
$$

As weights are drawn from an exponential distribution, $\operatorname{Var}(w^+) = \mathbb{E}[w^+]^2$, we have

$$
\begin{aligned}
\operatorname{Var}(z_k) &= n_e \mathbb{E}[w^+]^2 (2\operatorname{Var}(x) + \mathbb{E}[x]^2) + n_i \mathbb{E}[w^-]^2 (2\operatorname{Var}(x) + \mathbb{E}[x]^2) \\
&= n_e \mathbb{E}[w^+]^2 (\mathbb{E}[x^2] + \operatorname{Var}(x)) + n_i \mathbb{E}[w^-]^2 (\mathbb{E}[x^2] + \operatorname{Var}(x)) \\
&= (\mathbb{E}[x^2] + \operatorname{Var}(x))(n_e \mathbb{E}[w^+]^2 + n_i \mathbb{E}[w^-]^2) \\
&= \sigma_{\ell-1}^2 (\frac{2\pi - 1}{2\pi}) \mathbb{E}[w^+]^2 (n_e + \frac{n_e^2}{n_i})
\end{aligned}
\tag{24}
$$

Therefore $\mathbb{E}[w^+] = 1/(\frac{2\pi-1}{2\pi})(n_e + \frac{n_e^2}{n_i})$

Note that as the input to the network is all positive, the first weight matrix has no negative columns. We therefore use the bias vector to center the activations of the first layer (in other layers it is initialised to zeros).

$$
\mathbb{E}[z_k] = n_e \mathbb{E}[w^+]\mathbb{E}[x] + \beta_k
\tag{25}
$$

Therefore we initialise all elements of $\boldsymbol{\beta}$ to $-n_e \mathbb{E}[w^+]\mathbb{E}[x]$

## C.2 INITIALISATION OF DANN INHIBITORY WEIGHTS FOR BALANCED EXCITATION AND SUBTRACTIVE INHIBITION

Here provide details of inhibitory parameter initialisation such that $\mathbb{E}[z_k^E] = \mathbb{E}[(\mathbf{W}^{EI}\mathbf{z}^I)_k]$, for $\mathbf{W}^{EE} \overset{iid}{\sim} \operatorname{Exp}(\lambda^E)$.

$$
\begin{aligned}
\mathbb{E}[z_k^E] &= \mathbb{E}[\sum_i^d w_{ki}^{EE} x_i] = d\frac{1}{\lambda^E}\mathbb{E}[x] \\
\mathbb{E}[(\mathbf{W}^{EI}\mathbf{z}^I)_k] &= \mathbb{E}[\sum_j^{n_i} w_{kj}^{EI} \sum_i^d w_{ji}^{EI} x_i] = n_i \mathbb{E}[w^{EI}] d \mathbb{E}[w^{IE}]\mathbb{E}[x]
\end{aligned}
\tag{26}
$$

These expectaions are equal when both sets of excitatory weights are drawn from the same distribution, $\mathbf{W}^{IE} \overset{iid}{\sim} \operatorname{Exp}(\lambda^E)$ and $\mathbf{W}^{EI} \leftarrow \mathbb{1}/n_i$. Or alternatively, inhibitory weights can both drawn from the same distribution, $\mathbf{W}^{IE}, \mathbf{W}^{EI} \overset{iid}{\sim} \operatorname{Exp}(\sqrt{\lambda^E n_i})$. Note, that although the above always holds in expectation, in the case of a multiple inhibitory units we can apply the law of large numbers to conclude that the subtractive inhibition and excitatory input will be approximately equal.

Note that while this initialisation is general to different settings of $\lambda^E$, we initialise $\lambda^E \leftarrow \sqrt{d(2\pi - 1)}/\sqrt{2\pi}$ (see section D.1).

---

# D PROPORTIONAL RELATIONSHIP BETWEEN EXCITATORY INPUT MEAN AND STANDARD DEVIATION

Here we provide further details regarding the proportionality between $\mathbf{z}^{\mathrm{E}}$'s mean and standard deviation. This proportionality constant depends on which statistic or distribution that is of interest for activations (e.g. layer-statistics or unit batch-statistics as in layer and batch normalisation).

## D.1 UNIT STATISTICS OVER DATA AND PARAMETER DISTRIBUTIONS

As discussed in the main text, if we consider $c \cdot \mathbb{E}[z_k^{\mathrm{E}}] = \mathrm{Var}(z_k^{\mathrm{E}})^{1/2}$ for a unit $k$, with expectation over the data and parameters, $c = \sqrt{2\pi - 1}/\sqrt{d}$:

$$
\begin{aligned}
\mathbb{E}[z_k^{\mathrm{E}}] &= d \cdot \mathbb{E}[w^{\mathrm{EE}}]\mathbb{E}[x] \\
&= d \cdot \mathbb{E}[w^{\mathrm{EE}}]\frac{\sigma_{\ell-1}}{\sqrt{2\pi}} \\
\mathrm{Var}(z_k^{\mathrm{E}}) &= \mathrm{Var}(\sum_i^d w_{ki}^{\mathrm{EE}}x_i) \\
&= d \cdot \mathrm{Var}(w^{\mathrm{EE}}x) \\
&= d \cdot \mathrm{Var}(w^{\mathrm{EE}})\mathbb{E}[x^2] + d \cdot \mathrm{Var}(x)\mathbb{E}[w^{\mathrm{EE}}]^2 \\
&= d \cdot \mathrm{Var}(w^{\mathrm{EE}})(\mathbb{E}[x^2] + \mathrm{Var}(x)) \\
&= d \cdot \mathrm{Var}(w^{\mathrm{EE}})\sigma_{\ell-1}^2\frac{2\pi - 1}{2\pi}
\end{aligned}
\tag{27}
$$

Where $\mathbb{E}[w^{\mathrm{EE}}]^2 = \mathrm{Var}(w^{\mathrm{EE}})$ for weights drawn from an exponential distribution. Therefore

$$
\begin{aligned}
\mathbb{E}[z_k^{\mathrm{E}}] \cdot c &= \sqrt{\mathrm{Var}(z_k^{\mathrm{E}})} \\
c &= \frac{\sqrt{2\pi - 1}}{\sqrt{d}}
\end{aligned}
\tag{28}
$$

Additionally, we see that for $\mathrm{Var}(z_k^E) = \sigma_{\ell-1}^2$ the variance of the distribution that elements of $\mathbf{W}^{\mathrm{EE}}$ are drawn from should be

$$
\mathrm{Var}(w^{\mathrm{EE}}) = \frac{2\pi}{d \cdot (2\pi - 1)}
\tag{29}
$$

and so we can set $\lambda^{\mathrm{E}} \leftarrow \sqrt{d(2\pi - 1)}/\sqrt{2\pi}$, for $\mathrm{Var}(z_k^{\mathrm{E}}) = \sigma_{\ell-1}^2$.

## D.2 UNIT STATISTICS OVER THE DATA DISTRIBUTION

If instead we consider a unit k, with excitatory weights $\mathbf{w}_k^{\mathrm{EE}}$ and expectation and variance taken only over the data we have the approximation:

$$
\begin{aligned}
\mathbb{E}[z_k^{\mathrm{E}}] &= \mathbb{E}[x] \sum_i^d w_{ki}^{\mathrm{EE}} \\
&\approx d \cdot \mathbb{E}[x]\mathbb{E}[w^{\mathrm{EE}}] \\
&= d \cdot \frac{\sigma_{\ell-1}}{\sqrt{2\pi}}\mathbb{E}[w^{\mathrm{EE}}]
\end{aligned}
\tag{30}
$$

Likewise the variance over the data can be approximated as

$$
\begin{aligned}
\mathrm{Var}(z_k^{\mathrm{E}}) &= \mathrm{Var}(x) \sum_i^d (w_{ki}^{\mathrm{EE}})^2 \\
&\approx d \cdot \mathrm{Var}(x) \cdot \mathbb{E}[(w^{\mathrm{EE}})^2] \\
&= d \cdot \sigma_{\ell-1}^2\frac{\pi - 1}{2\pi} \cdot 2 \cdot \mathbb{E}[w^{\mathrm{EE}}]^2
\end{aligned}
\tag{31}
$$

Therefore

$$
\mathrm{E}_{x \sim p(x)}[z_k^{\mathrm{E}}] \cdot c = \sqrt{\mathrm{Var}_{x \sim p(x)}[z_k^{\mathrm{E}}]}
$$
$$
c \approx \frac{\sqrt{2\pi - 2}}{\sqrt{d}} \tag{32}
$$

### D.3 LAYER STATISTICS OVER THE DATA AND PARAMETER DISTRIBUTIONS

Alternatively we can consider the mean and standard deviation of the layer statistics $\mu_{\mathbf{z}^{\mathrm{E}}}$, $\sigma_{\mathbf{z}^{\mathrm{E}}}$ as calculated if one was to apply layer normalisation to $\mathbf{z}^{\mathrm{E}}$. Here again, these statistics are proportionally related, but with the constant $\sqrt{\pi}/\sqrt{d}$.

If we were to apply layer normalisation to $\mathbf{z}^{\mathrm{E}}$, the layer statistics would be as follows:

$$
\mathbf{z} = \frac{\mathbf{g}}{\sigma_{\mathbf{z}^{\mathrm{E}}}}(\mathbf{z}^{\mathrm{E}} - \mu_{\mathbf{z}^{\mathrm{E}}}) + \boldsymbol{\beta} \qquad \mu_{\mathbf{z}^{\mathrm{E}}} = \frac{1}{n_e}\sum_j^{n_e} z_j^{\mathrm{E}} = \frac{1}{n_e}\sum_j^{n_e} \mathbf{w}_{j,:}^{\mathrm{EE}}\mathbf{x} \qquad \sigma_{\mathbf{z}^{\mathrm{E}}}^2 = \frac{1}{n_e - 1}\sum_j^{n_e}(z_j^{\mathrm{E}} - \mu_{\mathbf{z}^{\mathrm{E}}})^2 \tag{33}
$$

We now derive the relationship that the expectation of layer statistics are proportionally related by $\mathbb{E}[\mu_{\mathbf{z}^{\mathrm{E}}}] \cdot \sqrt{\pi}/\sqrt{d} = \mathbb{E}(\sigma_{\mathbf{z}^{\mathrm{E}}}^2)^{1/2}$. The expectation of $\mathbb{E}[\mu_{\mathbf{z}^{\mathrm{E}}}]$ is straightforward:

$$
\mathbb{E}[\mu_{\mathbf{z}^{\mathrm{E}}}] = d \cdot \mathbb{E}[w^{\mathrm{EE}}] \cdot \mathbb{E}[x] \tag{34}
$$

Turning to the derivation of $\mathbb{E}[\sigma_{\mathbf{z}^{\mathrm{E}}}^2]$:

$$
\mathbb{E}[\sigma_{\mathbf{z}^{\mathrm{E}}}^2] = \mathbb{E}[\frac{1}{n_e - 1}\sum_i^{n_e}(z_i^{\mathrm{E}} - \mu_{\mathbf{z}^{\mathrm{E}}})^2] \tag{35}
$$

$$
= \frac{1}{n_e - 1}\sum_i^{n_e} \mathbb{E}[(\mathbf{w}_{i,:}^{\mathrm{EE}}\mathbf{x} - \frac{1}{n_e}\sum_j^{n_e} \mathbf{w}_{j,:}^{\mathrm{EE}}\mathbf{x})^2] \tag{36}
$$

$$
= \frac{1}{n_e - 1}\sum_i^{n_e} \mathbb{E}[(\hat{z}_i)^2] \tag{37}
$$

where we have defined $\hat{z}_i = \mathbf{w}_{i,:}^{\mathrm{EE}}\mathbf{x} - \frac{1}{n_e}\sum_j^{n_e} \mathbf{w}_{j,:}^{\mathrm{EE}}\mathbf{x}$. We can obtain $\mathbb{E}[(\hat{z}_i)^2]$ by deriving $\mathbb{E}[\hat{z}_i]$ and $\mathrm{Var}(\hat{z}_i)$. As

$$
\hat{w}_{ij} = w_{ij}^{\mathrm{EE}} - \frac{1}{n_e}\sum_k^{n_e} w_{kj}^{\mathrm{EE}} = \frac{1}{n_e}\sum_{k=1, k\neq i}^{n_e}(w_{ij}^{\mathrm{EE}} - w_{kj}^{\mathrm{EE}}) \tag{38}
$$

we see that $\mathbb{E}[\hat{w}_{ij}] = 0$, and therefore $\mathbb{E}[\hat{z}_i] = 0$. For the variance of $\mathrm{Var}(\hat{z}_i)$ we start with $\mathrm{Var}(\hat{w}_{ij})$.

$$
\mathrm{Var}(\hat{w}_{ij}) = \frac{1}{n_e^2}\mathrm{Var}(\sum_{k=1, k\neq i}^{n_e}(w_{ij}^{\mathrm{EE}} - w_{kj}^{\mathrm{EE}})) \tag{39}
$$

$$
= \frac{1}{n_e^2}(\sum_{k=1, k\neq i}^{n_e}\mathrm{Var}(w_{ij}^{\mathrm{EE}} - w_{kj}^{\mathrm{EE}}) + \sum_{\substack{k=1, k'=1 \\ k, k'\neq i, \, k\neq k'}}^{n_e}\mathrm{Cov}(w_{ij}^{\mathrm{EE}} - w_{kj}^{\mathrm{EE}}, w_{ij}^{\mathrm{EE}} - w_{k'j}^{\mathrm{EE}})) \tag{40}
$$

$$
= \frac{1}{n_e^2}((n_e - 1)2\mathrm{Var}(w^{\mathrm{EE}}) + (n_e - 1)(n_e - 2)\mathrm{Var}(w^{\mathrm{EE}})) \tag{41}
$$

$$
= \frac{(n_e - 1)}{n_e}\mathrm{Var}(w^{\mathrm{EE}}) \tag{42}
$$

For $i \leq n_e$ we calculate $\text{Var}(\hat{z}_i)$, keeping in mind that for $i \leq n_e, j \leq d$, $x_j$ are iid, and equation (38) shows that $\hat{w}_{ij}$ are iid in the $j$'th coordinate, so we see that

$$\text{Var}(\hat{z}_i) = \text{Var}(\sum_j^d \hat{w}_{ij} x_j) = d\text{Var}(\hat{w}x) \tag{43}$$

Remembering the values of $\mathbb{E}[\hat{w}], \text{Var}(\hat{w})$, that $\mathbb{E}[X^2] = \text{Var}(X) + \mathbb{E}[X]^2$, and for independent $X, Y$, $\text{Var}(XY) = \text{Var}(X)\text{Var}(Y) + \text{Var}(X)\mathbb{E}[Y]^2 + \text{Var}(Y)\mathbb{E}[X]^2$, we have

$$\text{Var}(\hat{z}_i) = d(\text{Var}(\hat{w})\text{Var}(x) + \text{Var}(\hat{w})\mathbb{E}[x]^2 + \text{Var}(x)\mathbb{E}[\hat{w}]^2) \tag{44}$$

$$= d(\text{Var}(\hat{w})\mathbb{E}[x^2] + \text{Var}(x)\mathbb{E}[\hat{w}]^2) = d\text{Var}(\hat{w})\mathbb{E}[x^2] \tag{45}$$

$$= \frac{d(n_e - 1)}{n_e}\text{Var}(w^{\text{EE}})\mathbb{E}[x^2] \tag{46}$$

Now putting these terms together we can derive $\mathbb{E}[\sigma^2_{\mathbf{z}^{\text{E}}}]$.

$$\mathbb{E}[\sigma^2_{\mathbf{z}^{\text{E}}}] = \frac{1}{n_e - 1}\sum_i^{n_e}\mathbb{E}[(\hat{z}_i)^2] \tag{47}$$

$$= \frac{1}{n_e - 1}\sum_i^{n_e}\text{Var}(\hat{z}_i) \tag{48}$$

$$= d \cdot \text{Var}(w^{\text{EE}}) \cdot \mathbb{E}[x^2] \tag{49}$$

Therefore returning to $\mathbb{E}[\mu_{\mathbf{z}^{\text{E}}}] \cdot c = \mathbb{E}(\sigma^2_{\mathbf{z}^{\text{E}}})^{1/2}$ and keeping in mind that the variance of an exponential random variable is it's mean squared,

$$c = \frac{(d \cdot \text{Var}(w^{\text{EE}})\mathbb{E}[x^2])^{1/2}}{d \cdot \mathbb{E}[w^{\text{EE}}] \cdot \mathbb{E}[x]} = \frac{\sqrt{\mathbb{E}[x^2]}}{\sqrt{d} \cdot \mathbb{E}[x]} \tag{50}$$

We have assumed that $x$ follows a rectified normal distribution. Therefore, $\mathbb{E}[x] = \frac{\sigma_{l-1}}{\sqrt{2\pi}}, \mathbb{E}[x^2] = \frac{\sigma^2_{l-1}}{2}$. Resulting in:

$$c = \frac{\sqrt{\pi}}{\sqrt{d}} \tag{51}$$

We note that for a DANN layer with a single inhibitory unit, $\mu_{\mathbf{z}^{\text{E}}} = z^{\text{I}}$ as $\mathbf{W}^{\text{IE}} \leftarrow \frac{1}{n_e}\sum_j^{n_e}\mathbf{w}^{\text{EE}}_{j,:}$, and $\mathbf{W}^{\text{EI}} \leftarrow \mathbb{1}$. Therefore DANN divisive inhibition, $\gamma$, can be made equivalent to layer standard deviation at initialisation in expectation if $e^\alpha \leftarrow c$. However, these calculations apply for the case of multiple interneuron if one makes the approximation $\mu_{\mathbf{z}^{\text{E}}} \approx (\mathbf{W}^{\text{EI}}\mathbf{W}^{\text{IE}}\mathbf{x})_i$ for any $i$.

# E   PARAMETER UPDATES AND FISHER INFORMATION MATRIX

## E.1   LAYER FISHER INFORMATION MATRIX

We view a layer's activation as parameterising a conditional distribution from the exponential family $P(\mathbf{y}|\mathbf{x}; \boldsymbol{\theta}) = P(\mathbf{y}|\mathbf{z})$, independent in each coordinate of $\mathbf{y}|\mathbf{z}$.

$$\log P(\mathbf{y}|\mathbf{x}; \boldsymbol{\theta}) = \frac{\mathbf{y} \cdot \mathbf{z} - \eta(\mathbf{z})}{\phi} + c(\mathbf{y}, \phi) \tag{52}$$

$$\mathbb{E}[\mathbf{y}|\mathbf{x}; \boldsymbol{\theta}] = f(\mathbf{z}) = \eta'(\mathbf{z}) \qquad \mathrm{Cov}(\mathbf{y}|\mathbf{x}; \boldsymbol{\theta}) = \mathrm{diag}(\phi f'(\mathbf{z})) \tag{53}$$

where $f(\mathbf{z})$ is the activation function of the layer, and $\phi$, $\eta$, $c$ define the particular distribution in the exponential family. Note we take $\eta'(\mathbf{z})$, $f'(\mathbf{z})$ to denote the $\frac{\partial \eta}{\partial \mathbf{z}}$, $\frac{\partial f}{\partial \mathbf{z}}$.

$F(\boldsymbol{\theta})$ is defined as:

$$F(\boldsymbol{\theta}) = \mathop{\mathbb{E}}_{\mathbf{x} \sim P(\mathbf{x}), \mathbf{y} \sim P(\mathbf{y}|\mathbf{x}; \boldsymbol{\theta})} \left[ \frac{\partial \log P(\mathbf{y}|\mathbf{x}; \boldsymbol{\theta})}{\partial \boldsymbol{\theta}} \frac{\partial \log P(\mathbf{y}|\mathbf{x}; \boldsymbol{\theta})}{\partial \boldsymbol{\theta}}^T \right] \tag{54}$$

As

$$\begin{aligned} \frac{\partial}{\partial \boldsymbol{\theta}} \log P(\mathbf{y}|\mathbf{x}; \boldsymbol{\theta}) &= \frac{\partial \mathbf{z}}{\partial \boldsymbol{\theta}} \frac{\partial}{\partial \mathbf{z}} \left[ \frac{\mathbf{y} \cdot \mathbf{z} - \eta(\mathbf{z})}{\phi} + c(\mathbf{y}, \phi) \right] \\ &= \frac{1}{\phi} \frac{\partial \mathbf{z}}{\partial \boldsymbol{\theta}} (\mathbf{y} - \frac{\partial \eta}{\partial \mathbf{z}}) \end{aligned} \tag{55}$$

we have

$$F(\boldsymbol{\theta}) = \mathop{\mathbb{E}}_{\mathbf{x} \sim P(\mathbf{x}), \mathbf{y} \sim P(\mathbf{y}|\mathbf{x}; \boldsymbol{\theta})} \left[ \frac{\partial \mathbf{z}}{\partial \boldsymbol{\theta}} \frac{(\mathbf{y} - \eta'(\mathbf{z}))}{\phi} \frac{(\mathbf{y} - \eta'(\mathbf{z}))^T}{\phi} \frac{\partial \mathbf{z}}{\partial \boldsymbol{\theta}}^T \right] \tag{56}$$

$$= \mathop{\mathbb{E}}_{\mathbf{x} \sim P(\mathbf{x})} \left[ \frac{\partial \mathbf{z}}{\partial \boldsymbol{\theta}} \mathop{\mathbb{E}}_{\mathbf{y} \sim P(\mathbf{y}|\mathbf{x}; \boldsymbol{\theta})} \left[ \frac{(\mathbf{y} - \eta'(\mathbf{z}))}{\phi} \frac{(\mathbf{y} - \eta'(\mathbf{z}))^T}{\phi} \Big| \mathbf{x}; \boldsymbol{\theta} \right] \frac{\partial \mathbf{z}}{\partial \boldsymbol{\theta}}^T \right] \tag{57}$$

$$= \mathop{\mathbb{E}}_{\mathbf{x} \sim P(\mathbf{x})} \left[ \frac{\partial \mathbf{z}}{\partial \boldsymbol{\theta}} \frac{\mathrm{Cov}\left[\mathbf{y}|\mathbf{x}; \boldsymbol{\theta}\right]}{\phi^2} \frac{\partial \mathbf{z}}{\partial \boldsymbol{\theta}}^T \right] \tag{58}$$

$$= \mathop{\mathbb{E}}_{\mathbf{x} \sim P(\mathbf{x})} \left[ \frac{\partial \mathbf{z}}{\partial \boldsymbol{\theta}} \frac{\mathrm{diag}(f'(\mathbf{z}))}{\phi} \frac{\partial \mathbf{z}}{\partial \boldsymbol{\theta}}^T \right] \tag{59}$$

where we recognise the covariance matrix is diagonal:

$$\mathrm{Cov}\left[\mathbf{y}|\mathbf{x}; \boldsymbol{\theta}\right] = \mathrm{diag}(\mathrm{Var}(y_1|\mathbf{x}; \boldsymbol{\theta}), ..., \mathrm{Var}(y_{n_e}|\mathbf{x}; \boldsymbol{\theta}))$$

To analyse the approximate KL divergence resulting from the simple case of perturbing individual parameters of a single-layer DANN, we only need to consider the diagonal entries of the Fisher.

$$\mathrm{D}_{\mathrm{KL}}\left[P_{\boldsymbol{\theta}} \,\|\, P_{\boldsymbol{\theta} + \boldsymbol{\delta}_{\tilde{\theta}}}\right] \approx \frac{1}{2} \boldsymbol{\delta}_{\tilde{\theta}}^T \mathop{\mathbb{E}}_{\mathbf{x} \sim P(\mathbf{x})} \left[ \frac{\partial \mathbf{z}}{\partial \boldsymbol{\theta}} \frac{\mathrm{diag}(f'(\mathbf{z}))}{\phi} \frac{\partial \mathbf{z}}{\partial \boldsymbol{\theta}}^T \right] \boldsymbol{\delta}_{\tilde{\theta}} \tag{60}$$

$$= \frac{\delta^2}{2\phi} \mathop{\mathbb{E}}_{\mathbf{x} \sim P(\mathbf{x})} \left[ \frac{\partial \mathbf{z}}{\partial \tilde{\theta}} \mathrm{diag}(f'(\mathbf{z})) \frac{\partial \mathbf{z}}{\partial \tilde{\theta}}^T \right] \tag{61}$$

$$= \frac{\delta^2}{2\phi} \mathop{\mathbb{E}}_{\mathbf{x} \sim P(\mathbf{x})} \left[ (f'(\mathbf{z})^T \odot \frac{\partial \mathbf{z}}{\partial \tilde{\theta}}) \frac{\partial \mathbf{z}}{\partial \tilde{\theta}}^T \right] \tag{62}$$

$$= \frac{\delta^2}{2\phi} \sum_k^{n_e} \mathop{\mathbb{E}}_{\mathbf{x} \sim P(\mathbf{x})} \left[ f'(z_k) \left(\frac{\partial z_k}{\partial \tilde{\theta}}\right)^2 \right] \tag{63}$$

where $\boldsymbol{\delta}_{\tilde{\theta}}$ represents a 1-hot vector corresponding to $\tilde{\theta}$, multiplied by a scalar $\delta$.

## E.2  DERIVATIVES

Here we provide derivatives for DANN layer activations with respect to the different parameter groups. The equations for the layer activation can be written

$$\mathbf{z} = \frac{\mathbf{g}}{\boldsymbol{\gamma}} \odot (\mathbf{z}^{\mathrm{E}} - \mathbf{W}^{\mathrm{EI}}\mathbf{z}^{\mathrm{I}}) + \boldsymbol{\beta}$$

$$\text{where} \quad \mathbf{z}^{\mathrm{E}} = \mathbf{W}^{\mathrm{EE}}\mathbf{x} \quad \mathbf{z}^{\mathrm{I}} = \mathbf{W}^{\mathrm{IE}}\mathbf{x} \quad \boldsymbol{\gamma} = \mathbf{W}^{\mathrm{EI}}(e^{\boldsymbol{\alpha}} \odot \mathbf{z}^{\mathrm{I}}) \tag{64}$$

Note $\dfrac{\partial z_k}{\partial w_{ij}^{\mathrm{EE}}} = \dfrac{\partial z_k}{\partial w_{ij}^{\mathrm{EI}}} = 0$ for $k \neq i$.

$$\frac{\partial z_i}{\partial w_{ij}^{\mathrm{EE}}} = \frac{\partial}{\partial w_{ij}^{\mathrm{EE}}} \left( \frac{g_i}{\gamma_i}(z_i^{\mathrm{E}} - (\mathbf{W}^{\mathrm{EI}}\mathbf{z}^{\mathrm{I}})_i) + \beta_i \right) \tag{65}$$

$$= \frac{g_i}{\gamma_i} \frac{\partial}{\partial w_{ij}^{\mathrm{EE}}}(z_i^{\mathrm{E}}) \tag{66}$$

$$= \frac{g_i}{\gamma_i} x_j \tag{67}$$

$$\frac{\partial z_i}{\partial w_{ij}^{\mathrm{EI}}} = \frac{\partial}{\partial w_{ij}^{\mathrm{EI}}} \left( \frac{g_i}{\gamma_i}(z_i^{\mathrm{E}} - (\mathbf{W}^{\mathrm{EI}}\mathbf{z}^{\mathrm{I}})_i) + \beta_i \right) \tag{68}$$

$$= -\frac{g_i}{\gamma_i^2} \frac{\partial \gamma_i}{\partial w_{ij}^{\mathrm{EI}}}(z_i^{\mathrm{E}} - (\mathbf{W}^{\mathrm{EI}}\mathbf{z}^{\mathrm{I}})_i) - \frac{g_i}{\gamma_i} z_j^{\mathrm{I}} \tag{69}$$

$$= -\frac{g_i}{\gamma_i^2} e^{\alpha_j} z_j^{\mathrm{I}}(z_i^{\mathrm{E}} - (\mathbf{W}^{\mathrm{EI}}\mathbf{z}^{\mathrm{I}})_i) - \frac{g_i}{\gamma_i} z_j^{\mathrm{I}} \tag{70}$$

$$= -\frac{g_i}{\gamma_i} z_j^{\mathrm{I}} \left( \frac{e^{\alpha_j}}{\gamma_i}(z_i^{\mathrm{E}} - (\mathbf{W}^{\mathrm{EI}}\mathbf{z}^{\mathrm{I}})_i) + 1 \right) \tag{71}$$

$$= -\sum_k^d \frac{g_i}{\gamma_i} w_{jk}^{\mathrm{IE}} x_k \left( \frac{e^{\alpha_j}}{\gamma_i}(z_i^{\mathrm{E}} - (\mathbf{W}^{\mathrm{EI}}\mathbf{z}^{\mathrm{I}})_i) + 1 \right) \tag{72}$$

In contrast $\dfrac{\partial z_k}{\partial w_{ij}^{\mathrm{EI}}}, \dfrac{\partial z_k}{\partial \alpha_j} \neq 0$ for $k \neq i$.

$$\frac{\partial z_k}{\partial w_{ij}^{\mathrm{IE}}} = \frac{\partial}{\partial w_{ij}^{\mathrm{IE}}} \left( \frac{g_k}{\gamma_k}(z_k^{\mathrm{E}} - (\mathbf{W}^{\mathrm{EI}}\mathbf{z}^{\mathrm{I}})_k) + \beta_k \right) \tag{73}$$

$$= -\frac{g_k}{\gamma_k^2} \frac{\partial \gamma_k}{\partial w_{ij}^{\mathrm{IE}}}(z_k^{\mathrm{E}} - (\mathbf{W}^{\mathrm{EI}}\mathbf{z}^{\mathrm{I}})_k) - \frac{g_k}{\gamma_k} w_{ki}^{\mathrm{EI}} x_j \tag{74}$$

$$= -\frac{g_k}{\gamma_k^2} e^{\alpha_i} w_{ki}^{\mathrm{EI}} x_j(z_k^{\mathrm{E}} - (\mathbf{W}^{\mathrm{EI}}\mathbf{z}^{\mathrm{I}})_k) - \frac{g_k}{\gamma_k} w_{ki}^{\mathrm{EI}} x_j \tag{75}$$

$$= -\frac{g_k}{\gamma_k} w_{ki}^{\mathrm{EI}} x_j \left( \frac{e^{\alpha_i}}{\gamma_k}(z_k^{\mathrm{E}} - (\mathbf{W}^{\mathrm{EI}}\mathbf{z}^{\mathrm{I}})_k) + 1 \right) \tag{76}$$

$$\frac{\partial z_i}{\partial \alpha_j} = \frac{\partial}{\partial \alpha_j}\Big(\frac{g_i}{\gamma_i}(z_i^{\mathrm{E}} - (\mathbf{W}^{\mathrm{EI}}\mathbf{z}^{\mathrm{I}})_i) + \beta_i\Big) \tag{77}$$

$$= -\frac{g_i}{\gamma_i^2}\frac{\partial \gamma_i}{\partial \alpha_j}(z_i^{\mathrm{E}} - (\mathbf{W}^{\mathrm{EI}}\mathbf{z}^{\mathrm{I}})_i) \tag{78}$$

$$= -\frac{g_i}{\gamma_i^2}w_{ij}^{\mathrm{EI}}e^{\alpha_j}z_j^{\mathrm{I}}(z_i^{\mathrm{E}} - (\mathbf{W}^{\mathrm{EI}}\mathbf{z}^{\mathrm{I}})_i) \tag{79}$$

$$= -\sum_k^d \frac{g_i}{\gamma_i}w_{ij}^{\mathrm{EI}}w_{jk}^{\mathrm{IE}}x_k\frac{e^{\alpha_j}}{\gamma_i}(z_i^{\mathrm{E}} - (\mathbf{W}^{\mathrm{EI}}\mathbf{z}^{\mathrm{I}})_i) \tag{80}$$

$$\frac{\partial z_i}{\partial g_i} = \frac{1}{\gamma_i}(z_i^{\mathrm{E}} - (\mathbf{W}^{\mathrm{EI}}\mathbf{z}^{\mathrm{I}})_i) \tag{81}$$

$$\frac{\partial z_i}{\partial b_i} = 1 \tag{82}$$

$$\tag{83}$$

### E.3 Approximate KL divergence for weight updates

If we consider an update to an element $ij$ of $\mathbf{W}^{\mathrm{EE}}$ the approximate KL divergence is

$$\begin{aligned} \mathrm{D}_{\mathrm{KL}}\big[P_{\boldsymbol{\theta}} \,\|\, P_{\boldsymbol{\theta}+\boldsymbol{\delta}_{\mathbf{W}_{ij}^{\mathrm{EE}}}}\big] &\approx \frac{\delta^2}{2\phi}\sum_k^{n_e}\mathop{\mathbb{E}}_{\mathbf{x}\sim P(\mathbf{x})}\left[f'(z_k)\big(\frac{\partial z_k}{\partial w_{ij}^{\mathrm{EE}}}\big)^2\right] \\ &= \frac{\delta^2}{2\phi}\mathop{\mathbb{E}}_{\mathbf{x}\sim P(\mathbf{x})}\left[f'(z_i)\big(\frac{g_i}{\gamma_i}x_j\big)^2\right] \end{aligned} \tag{84}$$

as $\frac{\partial z_k}{\partial w_{ij}^{\mathrm{EE}}} = 0$ for $k \neq i$.

In contrast, for an update to an element $ij$ of $\mathbf{W}^{\mathrm{IE}}$ we sum over $n_e$ terms, as $\frac{\partial z_k}{\partial w_{ij}^{\mathrm{IE}}} \neq 0$ for $k \neq i$.

$$\begin{aligned} \mathrm{D}_{\mathrm{KL}}\big[P_{\boldsymbol{\theta}} \,\|\, P_{\boldsymbol{\theta}+\boldsymbol{\delta}_{\mathbf{W}_{ij}^{\mathrm{IE}}}}\big] &\approx \frac{\delta^2}{2\phi}\sum_k^{n_e}\mathop{\mathbb{E}}_{\mathbf{x}\sim P(\mathbf{x})}\left[f'(z_k)\big(\frac{\partial z_k}{\partial w_{ij}^{\mathrm{IE}}}\big)^2\right] \\ &= \frac{\delta^2}{2\phi}\sum_k^{n_e}\mathop{\mathbb{E}}_{\mathbf{x}\sim P(\mathbf{x})}\left[f'(z_k)\big(-\frac{g_k}{\gamma_k}w_{ki}^{\mathrm{EI}}x_ja_{ki}\big)^2\right] \\ &= \frac{\delta^2}{2\phi}\sum_k^{n_e}\mathop{\mathbb{E}}_{\mathbf{x}\sim P(\mathbf{x})}\left[f'(z_k)\big(\frac{g_k}{\gamma_k}x_j\big)^2(w_{ki}^{\mathrm{EI}}a_{ki})^2\right] \end{aligned} \tag{85}$$

where $a_{kj} = \frac{e^{\alpha_j}}{\gamma_k}(z_k^{\mathrm{E}} - (\mathbf{W}^{\mathrm{EI}}z^{\mathrm{I}})_k) + 1$.

For an update $\boldsymbol{\delta}_{\mathbf{W}_{ij}^{\text{EI}}}$, while $\frac{\partial z_k}{\partial w_{ij}^{\text{EI}}} = 0$ for $k \neq i$, the derivative contains a $z_j^{\text{I}}$ term, so there is instead a squared sum over $d$ terms.

$$
\begin{aligned}
\text{D}_{\text{KL}}\big[P_{\boldsymbol{\theta}} \,\|\, P_{\boldsymbol{\theta}+\boldsymbol{\delta}_{\mathbf{W}_{ij}^{\text{EI}}}}\big] &\approx \frac{\delta^2}{2\phi} \sum_k^{n_e} \mathop{\mathbb{E}}_{\mathbf{x}\sim P(\mathbf{x})} \left[ f'(z_k) \big(\frac{\partial z_k}{\partial w_{ij}^{\text{EI}}}\big)^2 \right] \\
&= \frac{\delta^2}{2\phi} \mathop{\mathbb{E}}_{\mathbf{x}\sim P(\mathbf{x})} \left[ f'(z_i) \big(\frac{\partial z_i}{\partial w_{ij}^{\text{EI}}}\big)^2 \right] \\
&= \frac{\delta^2}{2\phi} \mathop{\mathbb{E}}_{\mathbf{x}\sim P(\mathbf{x})} \left[ f'(z_i) \big(- \frac{g_i}{\gamma_i} z_j^{\text{I}} a_{ij}\big)^2 \right] \\
&= \frac{\delta^2}{2\phi} \mathop{\mathbb{E}}_{\mathbf{x}\sim P(\mathbf{x})} \left[ f'(z_i) (z_j^{\text{I}})^2 (\frac{g_i}{\gamma_i})^2 (a_{ij})^2 \right] \\
&= \frac{\delta^2}{2\phi} \mathop{\mathbb{E}}_{\mathbf{x}\sim P(\mathbf{x})} \left[ f'(z_i) \big(\sum_n^d w_{j,n}^{\text{IE}} x_n\big)^2 (\frac{g_i}{\gamma_i})^2 (a_{ij})^2 \right] \\
&= \frac{\delta^2}{2\phi} \mathop{\mathbb{E}}_{\mathbf{x}\sim P(\mathbf{x})} \left[ f'(z_i) \big(\sum_n^d (w_{jn}^{\text{IE}})^2 (x_n)^2 + \sum_{n\neq m}^d w_{jn}^{\text{IE}} w_{jm}^{\text{IE}} x_n x_m\big) (\frac{g_i}{\gamma_i})^2 (a_{ij})^2 \right] \\
&= \frac{\delta^2}{2\phi} \sum_n^d \mathop{\mathbb{E}}_{\mathbf{x}\sim P(\mathbf{x})} \left[ f'(z_i) (w_{jn}^{\text{IE}})^2 (x_n)^2 (\frac{g_i}{\gamma_i})^2 (a_{ij})^2 \right] \\
&\quad + \frac{\delta^2}{2\phi} \sum_{n\neq m}^d \mathop{\mathbb{E}}_{\mathbf{x}\sim P(\mathbf{x})} \left[ f'(z_i) w_{jn}^{\text{IE}} w_{jm}^{\text{IE}} x_n x_m (\frac{g_i}{\gamma_i})^2 (a_{ij})^2 \right]
\end{aligned}
\tag{86}
$$

Finally, for alpha

$$
\begin{aligned}
\text{D}_{\text{KL}}\big[P_{\boldsymbol{\theta}} \,\|\, P_{\boldsymbol{\theta}+\boldsymbol{\delta}_{\alpha_i}}\big] &\approx \frac{\delta^2}{2\phi} \sum_k^{n_e} \mathop{\mathbb{E}}_{\mathbf{x}\sim P(\mathbf{x})} \left[ f'(z_k) \big(\frac{\partial z_k}{\partial \alpha_i}\big)^2 \right] \\
&= \frac{\delta^2}{2\phi} \sum_k^{n_e} \mathop{\mathbb{E}}_{\mathbf{x}\sim P(\mathbf{x})} \left[ f'(z_k) \big(- \sum_j^d \frac{g_k}{\gamma_k} w_{ki}^{\text{EI}} w_{ij}^{\text{IE}} x_j \frac{e^{\alpha_i}}{\gamma_k} (z_k^{\text{E}} - (\mathbf{W}^{\text{EI}}\mathbf{z}^{\text{I}})_k)\big)^2 \right] \\
&= \frac{\delta^2}{2\phi} \sum_k^{n_e} \mathop{\mathbb{E}}_{\mathbf{x}\sim P(\mathbf{x})} \left[ f'(z_k) \big(- \sum_j^d \frac{g_k}{\gamma_k} w_{ki}^{\text{EI}} w_{ij}^{\text{IE}} x_j (a_{ki} - 1)\big)^2 \right] \\
&= \frac{\delta^2}{2\phi} \sum_k^{n_e} \mathop{\mathbb{E}}_{\mathbf{x}\sim P(\mathbf{x})} \left[ f'(z_k) \big(\sum_j^d (\frac{g_k}{\gamma_k} w_{ki}^{\text{EI}} w_{ij}^{\text{IE}} x_j (a_{ki} - 1))^2 \right. \\
&\qquad\qquad\qquad\qquad \left. + \sum_{n\neq m}^d ((\frac{g_k}{\gamma_k} w_{ki}^{\text{EI}})^2 w_{in}^{\text{IE}} x_n w_{im}^{\text{IE}} x_m (a_{ki} - 1)^2)\big) \right] \\
&= \frac{\delta^2}{2\phi} \sum_k^{n_e} \sum_j^d \mathop{\mathbb{E}}_{\mathbf{x}\sim P(\mathbf{x})} \left[ f'(z_k) (\frac{g_k}{\gamma_k} w_{ki}^{\text{EI}} w_{ij}^{\text{IE}} x_j (a_{ki} - 1))^2 \right] \\
&\quad + \frac{\delta^2}{2\phi} \sum_k^{n_e} \sum_{n\neq m}^d \mathop{\mathbb{E}}_{\mathbf{x}\sim P(\mathbf{x})} \left[ f'(z_k) (\frac{g_k}{\gamma_k} w_{ki}^{\text{EI}})^2 w_{in}^{\text{IE}} x_n w_{im}^{\text{IE}} x_m (a_{ki} - 1)^2 \right]
\end{aligned}
\tag{87}
$$

## F  ALGORITHMS

Here we provide pseudo-code for implementation details. Please see the following link for code: *https://github.com/linclab/ltlwdp*

### F.1  PARAMETER INITIALIZATION

---
**Algorithm 1** Parameter initialization for DANNs
---
    **for** layer $L$ **do**
        **require** $n_e, n_i, d$
        $\mathbf{W}^{\mathrm{EE}} \sim exp(\lambda^{\mathrm{E}})$
        **if** $n_i = 1$
            $\mathbf{W}^{\mathrm{IE}} \leftarrow \frac{1}{n_e} \sum_{j=1}^{n_e} \mathbf{w}_j^{\mathrm{EE}}$
            $\mathbf{W}^{\mathrm{EI}} \leftarrow \mathbb{1}$
        **else**:
            $\mathbf{W}^{\mathrm{IE}} \sim exp(\lambda^{\mathrm{E}})$
            $\mathbf{W}^{\mathrm{EI}} \leftarrow \mathbb{1}/n_i$
        **end if**
        $\boldsymbol{\alpha} \leftarrow \mathbb{1} \cdot log(\frac{\sqrt{2\pi-1}}{\sqrt{d}})$
        $\mathbf{g}, \boldsymbol{\beta} \leftarrow \mathbb{1}$
    **end for**
---

Where number of excitatory output units is $n_e$, number of inhibitory units $n_i$, and input dimensionality $d$ and $\lambda^{\mathrm{E}} = \sqrt{d(2\pi - 1)}/\sqrt{2\pi}$.

### F.2  PARAMETER UPDATES

For DANN parameter updates we used the algorithms detailed below. Note that gradients were corrected as detailed in Section 4 and see Algorithm 3.

All below algorithms are computed using the loss gradients $\nabla\boldsymbol{\theta}$ of parameter $\boldsymbol{\theta}$, in a given model computed on a minibatch sample.

---
**Algorithm 2** Parameter updates
---
    **Require** learning rate $\eta$, updates $\Delta\theta$
    **for** each layer $l$ **do**
        $\mathbf{W}^{\mathrm{EE}} \leftarrow \mathbf{W}^{\mathrm{EE}} - \eta\Delta\mathbf{W}^{\mathrm{EE}}$
        $\mathbf{W}^{\mathrm{IE}} \leftarrow \mathbf{W}^{\mathrm{IE}} - \eta\Delta\mathbf{W}^{\mathrm{IE}}$
        $\mathbf{W}^{\mathrm{EI}} \leftarrow \mathbf{W}^{\mathrm{EI}} - \eta\Delta\mathbf{W}^{\mathrm{EI}}$
        $\alpha \leftarrow \alpha - \eta\Delta\alpha$
        $\mathbf{g} \leftarrow \mathbf{g} - \eta\Delta\mathbf{g}$
        $\boldsymbol{\beta} \leftarrow \boldsymbol{\beta} - \eta\Delta\boldsymbol{\beta}$
        $\mathbf{W}^{\mathrm{EE}} \leftarrow max(\mathbf{W}^{\mathrm{EE}}, 0)$
        $\mathbf{W}^{\mathrm{IE}} \leftarrow max(\mathbf{W}^{\mathrm{IE}}, 0)$
        $\mathbf{W}^{\mathrm{EI}} \leftarrow max(\mathbf{W}^{\mathrm{EI}}, 0)$
        $\mathbf{g} \leftarrow max(\mathbf{g}, 0)$
    **end for**
---

### F.3  DANN GRADIENT CORRECTION ALGORITHMS

For the majority of experiments we scaled gradients using the heuristic correction terms derived in Section 4 (and see Appendix E). In this case we applied the following algorithm before Algorithm 2.

---

**Algorithm 3** DANN gradient correction

---

**for** each layer $l$ **do**
    **require** $n_e, n_i, d$
    $\Delta \mathbf{W}^{\mathrm{EE}} \leftarrow \nabla \mathbf{W}^{\mathrm{EE}}$
    $\Delta \mathbf{W}^{\mathrm{IE}} \leftarrow \frac{1}{\sqrt{n_e}} \nabla \mathbf{W}^{\mathrm{IE}}$
    $\Delta \mathbf{W}^{\mathrm{EI}} \leftarrow \frac{1}{d} \nabla \mathbf{W}^{\mathrm{EI}}$
    $\Delta \boldsymbol{\alpha} \leftarrow \frac{1}{d\sqrt{n_e}} \nabla \boldsymbol{\alpha}$
    $\Delta \mathbf{g} \leftarrow \nabla \mathbf{g}$
    $\Delta \boldsymbol{\beta} \leftarrow \nabla \boldsymbol{\beta}$
**end for**

---

We also tested that our heuristic correction factors approximated gradient multiplication by the diagonal of $F_t^{-1}$ for each layer (see Figure 2).

---

**Algorithm 4** Gradient correction by approximation of $\mathrm{diag}(\mathbf{F}^{-1})$

---

**Require** learning rate $\eta$, fisher momentum $k$, fisher learning rate $\lambda$
**for** each batch $(x_t, y_t)$ **do**
    Compute $p = \mathrm{softmax}(z)$
    Compute cross entropy loss $L(p, y_t)$
    **for** each layer $l$ **do**
        $\nabla \boldsymbol{\theta}^l \leftarrow \frac{\partial L}{\partial \theta^l}$
    **end for**
    Sample $\hat{y} \sim \mathrm{Categorical(p)}$
    Compute cross entropy loss $L(p, \hat{y})$
    **for** each layer $l$ **do**
        $\hat{F} = \frac{1}{\lambda \cdot |batch|} \sum_i^{|batch|} (\frac{\partial L}{\partial \theta^l})_i^2$
        $F_t = k\hat{F} + (1-k)F_{t-1}$
        $F_t^{-1} = 1/F_t$
        $F_t^* = F_t^{-1} \cdot 1/||F_t^{\mathbf{W}^{\mathrm{EE}}}||$ where $F^{\mathbf{W}^{\mathrm{EE}}}$ is the elements of $F$ corresponding to $\mathbf{W}^{\mathrm{EE}}$
        $\Delta \boldsymbol{\theta}^l \leftarrow F_t^* \nabla \boldsymbol{\theta}^l$
    **end for**
**end for**

---

Here we note this update can be considered very rough diagonal approximation to natural gradient descent. In addition, various efficient approximations to natural gradient descent that have been utilized such as KFAC Martens & Grosse (2015) could not be applied due to the structure of DANNs, as the mathematical assumptions of KFAC, which were made for feedforward networks with activations as matrix multiplications, do not apply.

## F.4   Learning rate scaling and gradient normalisation

We also tested whether constraining the gradient norm and scaling the learning rate based on parameter clipping improved DANN performance. For these experiments we applied the following algorithms.

---

**Algorithm 5** Gradient normalisation

---

**for** each layer $\ell$ **do**
    **Require** $\nabla \theta^\ell, M$
    **if** $||\nabla \theta^\ell||_2 > M$:
        $\Delta \theta^\ell \leftarrow M \cdot \frac{\nabla \theta^\ell}{||\nabla \theta^\ell||_2}$
    **else**:
        $\Delta \theta^\ell \leftarrow \nabla \theta^\ell$
    **end if**
**end for**

---

---

**Algorithm 6** Learning rate scaling

---

    **for** each layer $\ell$ **do**
        **Require** $\nabla\theta^\ell, M, \boldsymbol{\xi}$
        $i \leftarrow 1$
        $c \leftarrow 0$
        **while** $c < M$:
            $\eta \leftarrow \xi_i$
            $c \leftarrow \text{CosineSimilarity}(\max(0, \theta^\ell - \eta\nabla\theta^\ell), \theta^\ell - \eta\nabla\theta^\ell)$
            $i \leftarrow i + 1$
    **end for**

---

The learning rate scaling method temporarily reduces the learning rate whenever parameter clipping would reduce the cosine of the angle, made between the gradient and actual updates, below a certain constraint. For any optimization problems caused by actual clipped updates not following the gradient, learning rate scaling is a principled way of following the direction of the gradient.

We also note, this technique can be generally applied to any other model which is constrained so that it cannot have updates freely follow gradient descent. If the constrained parameter space is an open subset of euclidean space, and we allow the learning rate to be arbitrarily small (Algorithm 6 with $\lim_{i\to\infty} \xi_i = 0$), updates will always follow the direction of the gradient.

