# OpenReview forum: "Learning to live with Dale's principle: ANNs with separate excitatory and inhibitory units"
_ICLR.cc/2021/Conference — ICLR 2021 Poster_

### Official Review · AnonReviewer1 · 2020-10-26
**Gain modulation of inhibitory feedforward inhibition**

**Rating:** 9
**Confidence:** 5

**Review:**

This is a great investigation on how to scale the gain of the inhibitory weights to balance the impact that the changes that the excitatory and inhibitory connections have on the layer’s output. I think using the KL distance that naturally connects with the Fisher Information is neat. I appreciate the effort that the authors make to connect the manner neural circuits are designed and connect it with ANN. You never know when the breakthrough can arise.

I love the experiments that the authors present illustrating with clarity the impact that having the proper gain modulation of the inhibitory changes have in the speed of convergence.

My single constructive criticism is that the inspiration in cortical circuits do not prevent the authors to get inspiration from smaller neural circuits like in insects for example. The Mushroom Bodies of the insects are the equivalent of the cortex and present feedforward inhibition. The number of layers is much smaller but the neural principles that operate are fairly consistent across multiple animal species. Drawing from that experience, the mutual inhibition within layer may provide a natural mechanism to keep balance in the output distribution as shown for example in mean field models that investigate the regulation of activity in a dynamical neural layer (see for example https://journals.plos.org/ploscompbiol/article?id=10.1371/journal.pcbi.1003133).

Other that this comment I learn and enjoy from reading this paper. I think it should be accepted.

---

> ### Author Response · Authors · 2020-11-18
> **Checking our cortical chauvinism**
>
> We thank the reviewer for their kind and constructive comments. We fully agree that non-cortical circuits can provide equal inspiration for these investigations and we will specifically add statements and references to that effect, e.g. noting the mushroom body of insects. Moreover, we will discuss the interesting observation that there may be very general principles at play with respect to maintaining balanced output via mutual inhibition, reference the paper the reviewer noted, and propose this as a future extension of our work.

---

### Official Review · AnonReviewer3 · 2020-10-27
**Dale's principle may not reduce the performance of feedforward ANNs if one uses negative weights only for feedforward inhibition.**

**Rating:** 6
**Confidence:** 4

**Review:**

Summary: It is shown that Dale’s principle can be observed in feedfoward ANNs if one uses inhibitory neurons in the form of feedforward inhibition, while the other neurons are purely excitatory.

Pros: This is a nice and new insight. It appears to be useful for understanding the design of biological neural networks, and at least one type of uses of inhibitory neurons in them.

Cons:  Apparently this insight provides no benefit for designing ANN. Furthermore the biological insight is rather limited because biological neural networks are not feedforward networks. Also, the chosen tasks (3 variations of MNIST) are relatively simple, and are solved with relatively shallow networks, with just 4 hidden layers. In my view this evaluation does not support the much more general claim in the Abstract that „ANN’s that respect Dale’s principle can be built without sacrificing learning performance“.

---

> ### Author Response · Authors · 2020-11-18
> **Pushing the performance of Dale’s ANNs**
>
> We are very happy that the reviewer found our paper insightful, and we thank them for their constructive critiques. Our responses are as follows:
>
> *1) Apparently this insight provides no benefit for designing ANN.*
>
> Indeed, as the reviewer notes here, we did not observe better performance with DANNs than with standard ANNs. Of course, the goal of this paper was to close the gap in learning performance between standard ANNs and ANNs that obey Dale’s principle. One reason that this is important is simply that ANNs that obey Dale’s principle, but which are not impaired at learning relative to normal ANNs, will be a useful tool for neuroscience research. However, we also wonder about potential computational benefits to Dale’s principle, and hope in future work to explore this possibility. We see this paper, which closes the learning gap, as a key initial step towards these future investigations. Please see also our reply to Reviewer 2, comment 2 for more discussion on this matter.
>
> *2) Furthermore the biological insight is rather limited because biological neural networks are not feedforward networks.*
>
> The reviewer is correct that real neural circuits are typically recurrent, and thus, it would be beneficial to also consider how DANNs can operate within the recurrent context. However, thanks to the mathematical similarity between a multilayer feedforward neural network and a recurrent neural rolled out through time (see e.g. Liao and Poggio, 2016, https://arxiv.org/abs/1604.03640), making this connection is relatively straightforward. In fact, our formulation of DANNs is fully applicable to recurrent neural networks, thanks to these connections. We propose to add a section to the appendix describing how our formulations can be ported to the case of recurrent neural networks. If the reviewers agree that this is a good idea, we will include this in our revised manuscript. Please see also our response to Reviewer 2, comment 1.
>
> *3) Also, the chosen tasks (3 variations of MNIST) are relatively simple, and are solved with relatively shallow networks, with just 4 hidden layers. In my view this evaluation does not support the much more general claim in the Abstract that „ANN’s that respect Dale’s principle can be built without sacrificing learning performance“.*
>
> We agree that the tasks we explored here were relatively simple. However, as noted by Reviewer 4, despite these tasks being simple this is, to our knowledge, the first paper to show ANNs that obey Dale’s principle that can learn on these tasks as well as standard ANNs. Nonetheless, to expand on our results, we have been exploring the use of DANN style architectures in deep convolutional networks. First we note that the response of a convolutional network can be expressed as a normal matrix multiplication where the rows of the weight matrix correspond to convolutional filters, and the columns of the input matrix correspond to the different filter locations. As such, we can readily express the same DANN formulation for convolutional networks. Second, we have preliminary results showing that learning in DANN convnets is approximately as good as learning in regular convnets (see the attached figure 2 in https://pdfhost.io/v/AFBEMscCX_DANN_Preliminary_Responsepdf.pdf). We propose to add discussion of how to apply the DANN formalism to convnets, and if the reviewers feels it is important, we can include a more thorough version of this data (after appropriate hyperparameter optimization) in the final version of the paper. Though, we note that full hyperparameter optimization and experimentation will take some time, so the final results of these experiments may not be ready by next week.

---

### Official Review · AnonReviewer4 · 2020-10-28
**An interesting submission on training ANNs with E/I neuronal division**

**Rating:** 6
**Confidence:** 4

**Review:**

Most neurons in the brains are either excitatory (E) or inhibitory (I) - sometimes referred to as Dale’s law.  Practically Dale’s principle is often left out of Artificial Neural Networks (ANNs) because having the E and I separation often impairs learning, although this has not been well documented in the literature (probably due to that this is also interpreted as a negative result). In this paper, the authors propose a new scheme to construct and train the feedforward E/I network by incorporating several ingredients, including feedforward inhibition and E/I balance among others. It is shown that this particular kind of E/I networks (DANNs) trained on MNIST and variations of MNIST could achieve a level of performance that is comparable to those without E/I separation.

Quality: I think this is an interesting submission of good quality, with some novel ideas and promising preliminary results.
Clarity: The writing is generally clear.
Originality: As far as I can tell, the results are original.
Significance: Although the results are promising, I have reservations about the significance of these results as the performance of the models are still worst than the standard ANNs.

Pros:
1.To my knowledge, this is the first E/I network that could achieve comparable performance with the standard ANN model on MNIST task (although at the same time, I have to say that not too many papers have studied and reported this issue).
2. The ingredients in the proposed model is well motivated in neuroscience, such as the feedforward inhibition, and E/I balance, as well as no connections between I neurons across the different layers.
3.   The results on the MNIST and its variations look promising.
4.  The paper is fairly well written and the basic ideas are clear.

Cons:
1.The role of the subtractive and divisive components need to be better explained. Are both of them necessary for getting the results shown later?
2. The authors assume the number of E neurons is far larger than that of the I neurons. This is not quite true in physiology. The E/I ratio reported is often around 4:1. The authors assumed 10% of neurons are I neurons- this is on the smaller end. Another related concern is that, in cortex, despite of a smaller number, I neurons are often responsible for controlling the dynamics/computation due to the dense connectivity from I to E neurons. I am a little bit worried that the paper is studying a quite different regime, in which the E neurons are dominating. Also, would adding more I neurons decrease the performance of the network? If that is the case, that would be concerning.
3. The initialization of E/I network has been carefully studied previously in the context of training balanced E/I recurrent neural networks (e.g., Ingrosso & Abbott, 2019, which the authors cited). How does the authors scheme different from the previous work?
4. The method assumes inhibitory units are linear units. Several questions arise. First, is this a mathematical issue or a numerical issues? Second, does this imply the firing rate of inhibitory neuron can be both positive and negative?
5. In fig4, DANN performs significantly worse than LayerNorm and BathNorm.
6.The algorithms is not tested on slightly more challenging benchmark datasets such as CIFAR10 or ImageNet. Relatedly, would DANN scale up to larger networks?

Questions to be clarified:
*Are their connections between the I neurons within the same layers?
*page 4, “Unlike a column constrained network, a layer in a DANN is not restricted in its potential function space. “ - It is unclear what this sentence means…
*Between Eq 4 and Eq 5, the authors mentioned the exponential family. What particular distribution was used? Gaussian or any exponential family distribution would produce similar results?
*The authors wrote: “As a result, inhibitory unit parameters updates are scaled down relative to excitatory parameter updates. This is intriguing given the differences between inhibitory and excitatory neuron plasticity…including the relative extent of weight changes in excitatory and inhibitory neurons (McBain et al., 1999). ” I think these comparisons to the neuroscience literature are too vague and potentially mis-leading. To make this useful, it would helpful to make the comparison more specific and clear.
*I am worried that the experiments for the ColumnEi model was not treated fairly. In section 5.1, it is mentioned that 50 columns are negative. Did the authors try to make increase this number to see if the performance would be improved for the ColumnEi model?


*********updated after rebuttal period
I still consider this as an interesting contribution, and stand with my original rating.
It would be useful if the discrepancies and similarity between the connectivity structures in the model and the anatomy could be more carefully discussed in the paper.

---

> ### Author Response · Authors · 2020-11-18
> **Clarifying and expanding on several points 2/2**
>
> *5) In fig4, DANN performs significantly worse than LayerNorm and BathNorm.*
>
> We would argue that figure 4 shows equivalent performance on K-MNIST. But, the reviewer is correct that the DANN performance is not quite as good as LayerNorm and BatchNorm on Fashion-MNIST. However, we would note that they are actually quite close. For example, if the reviewer looks at Table 3, they will see that on the test set DANNs achieved an error rate of 10.962 +/- 0.365, compared to 10.445 +/- 0.455 for LayerNorm and 9.992 +/- 0.218 for BatchNorm, which is a real difference, but arguably not huge. For comparison, the column constrained ANN achieved only 14.986 +/- 0.674. Moreover, we would note that the DANN performance was within the standard deviation of the MLP performance.
>
> *6)The algorithms is not tested on slightly more challenging benchmark datasets such as CIFAR10 or ImageNet. Relatedly, would DANN scale up to larger networks?*
>
> While, to our knowledge, this is the first paper to show ANNs that obey Dale’s principle learning as well as standard ANNs on simple tasks, this is a very important question. In-line with our response to Reviewer 3, comment 3, we note that the corrections derived in the paper apply to convolutional networks, and we have been running experiments on deep convolutional DANNs trained on CIFAR-10. We have preliminary data (see attached figure 2 in https://pdfhost.io/v/AFBEMscCX_DANN_Preliminary_Responsepdf.pdf) suggesting that DANN convnets have performance approximately equal to that of standard convnets on this dataset. We intend on running a more thorough set of experiments following on this preliminary data (with full hyperparameter tuning), though this may take some time. We propose to add discussion of how to apply the DANN formalism to convnets, and if the reviewers feel that it is important, we can include the results of these experiments in the camera ready version of the manuscript.
>
> *7) Are their connections between the I neurons within the same layers?*
>
> No, there are not. We will clarify this in the paper.
>
> *8) page 4, “Unlike a column constrained network, a layer in a DANN is not restricted in its potential function space. “ - It is unclear what this sentence means…*
>
> We can see how this sentence is unclear. What we mean by this is that in a column constrained network there are literally many functions that a single layer cannot approximate, because the linear operation is constrained to matrices with columns that have only positive or negative signs. In contrast, in DANNs, the initial linear integration in the excitatory units can match any linear function. We propose expanding this sentence to clarify this point.
>
> *9) Between Eq 4 and Eq 5, the authors mentioned the exponential family. What particular distribution was used? Gaussian or any exponential family distribution would produce similar results?*
>
> Our mathematical analysis only assumes any distribution from the natural exponential family (the exponential family with T(y) = y, see footnote 1). So, it applies equally to any such distribution. This group of distributions includes the Gaussian, Poisson, gamma, and binomial distributions. We propose to clarify this point in the footnote.
>
> *10) The authors wrote: “As a result, inhibitory unit parameters updates are scaled down relative to excitatory parameter updates. This is intriguing given the differences between inhibitory and excitatory neuron plasticity…including the relative extent of weight changes in excitatory and inhibitory neurons (McBain et al., 1999). ” I think these comparisons to the neuroscience literature are too vague and potentially mis-leading. To make this useful, it would helpful to make the comparison more specific and clear.*
>
> This is a fair point. What we were referring to, ultimately, was the fact that inhibitory plasticity is fairly difficult to achieve experimentally. Indeed, in the past, many neuroscientists thought that it did not exist (such as McBain et al, 1999). Thus, all we intended to refer to here was the apparent reduced plasticity at inhibitory synapses in the brain. We will clarify this in this section.
>
> *11) I am worried that the experiments for the ColumnEi model was not treated fairly. In section 5.1, it is mentioned that 50 columns are negative. Did the authors try to make increase this number to see if the performance would be improved for the ColumnEi model?*
>
> This is an important point to clarify. To test this question, we ran additional experiments with ColumnEi models that contain 100 negative columns. We find that these models learn just as poorly as the other ColumnEi models. Please see the attached figure 1, table 1 in  https://pdfhost.io/v/AFBEMscCX_DANN_Preliminary_Responsepdf.pdf. We will include these results in the revised paper.

---

> ### Author Response · Authors · 2020-11-18
> **Clarifying and expanding on several points 1/2**
>
> We are very grateful to the reviewer for their thorough, fair and insightful comments on our paper. Here are our specific responses to the individual comments:
>
> *1) The role of the subtractive and divisive components need to be better explained. Are both of them necessary for getting the results shown later?*
>
> This is a very interesting question. We can say with certainty that the subtractive component is critical, as it provides the ability to match the function approximation capabilities of a normal ANN. With respect to the divisive component, it may be less necessary, but still useful. Specifically, the divisive component is initialized to provide some of the same benefits as other normalization schemes. Notably, other normalization approaches help learning and appear to make the system more robust to initialization (e.g. Zhang et al. 2019, https://arxiv.org/pdf/1901.09321.pdf). But, normalisation is not necessary for learning, per se. Informally, we have observed similar properties with divisive inhibition in our model. However, we should note that in our model the specific equivalence to normalization is not enforced after initialization. Future work could examine whether additional advantages could be drawn from developing techniques for ensuring continued equivalence to existing normalization schemes (as mentioned in the Discussion). We propose to add more discussion of this matter to the manuscript.
>
> *2) The authors assume the number of E neurons is far larger than that of the I neurons. This is not quite true in physiology. The E/I ratio reported is often around 4:1. The authors assumed 10% of neurons are I neurons- this is on the smaller end. Another related concern is that, in cortex, despite of a smaller number, I neurons are often responsible for controlling the dynamics/computation due to the dense connectivity from I to E neurons. I am a little bit worried that the paper is studying a quite different regime, in which the E neurons are dominating. Also, would adding more I neurons decrease the performance of the network? If that is the case, that would be concerning.*
>
> This is a good question, indeed, the reviewer is correct that the percentage of inhibitory neurons in cortical circuits is likely above 10%. We were simply being conservative in our choice of this number. But, it is important to ensure that our choice is not critical to our results. Given this, we have run new simulations with larger numbers of inhibitory neurons, such that the ratio is 4:1. We find that learning is in fact slightly better in this scenario (see attached figure 1, table 1 in https://pdfhost.io/v/AFBEMscCX_DANN_Preliminary_Responsepdf.pdf). We propose adding these results to the Appendix of the manuscript.
>
> *3) The initialization of E/I network has been carefully studied previously in the context of training balanced E/I recurrent neural networks (e.g., Ingrosso & Abbott, 2019, which the authors cited). How does the authors scheme different from the previous work*
>
> This is an excellent question. The Ingrosso & Abbott (2019) paper is indeed closely related to our paper. However, the Ingrosso & Abbott paper is focussed on two issues: (1) the development of an alternative to recursive least squares training for networks with separate excitatory and inhibitory units, (2) the development of networks that maintain “dynamic” balance (meaning the E/I balance occurs without synaptic updates) versus “parametric” balance (wherein E/I balance requires synaptic plasticity). Their goal was not to design techniques for training ANNs with gradient descent, nor was it to develop networks that obey Dale’s principle but which match the learning performance of traditional ANNs. Thus, the goals of the two papers, though related, are ultimately different. We propose to add some discussion of these differences, as well as the differences to other related papers, to the introduction in order to clarify the unique contributions of our work.
>
> *4) The method assumes inhibitory units are linear units. Several questions arise. First, is this a mathematical issue or a numerical issues? Second, does this imply the firing rate of inhibitory neuron can be both positive and negative?*
>
> These are important points to clarify. On the first, the choice was largely driven by ease of mathematical analysis. On the second, in our specific models, inhibitory neurons cannot have negative firing rates despite being linear. The reason is that they only receive positive inputs from the layer below and they do not have a bias term. Thus, their output is strictly non-negative. We propose to clarify this point in the manuscript. In future work that incorporates inhibitory inputs to the inhibitory units, we believe that using appropriate nonlinear functions for the inhibitory units will be important.

---

> > ### Comment · AnonReviewer4 · 2020-11-25
> > **further points**
> >
> > I thank the authors for addressing  my review. I have more points regarding the connection to I neurons.
> > My understanding is that, in the brain, I neurons mostly receive recurrent connections from the local circuits, not inputs from another brain area. And also I neurons typically do not project to another region. I neurons certainly can interact with other I neurons.
> >
> > Based on the authors reply, it appeared that in the proposed model, I neurons only receive inputs from the level below and there is no recurrent between the I neurons. Also, I neurons do not project to the next layer in the model, right?
> >
> > If my understanding is correct, these discrepancies between the connectivity structure in the model and the anatomy should be carefully stated and discussed in the paper.

---

> > > ### Author Response · Authors · 2020-11-25
> > > **response to further points**
> > >
> > > Thank you for your additional response. Yes, you are correct, in our model I neurons only project within layer and only receive input from the layer below, which is inspired by feedforward inhibition in the brain. However, we have also added a section to discuss how our results apply to feedback inhibition and recurrent networks in the appendix section B. Please also see our response to reviewer 2, point 1) for further discussion of this issue.
> > >
> > > As for I neuron connectivity, we would point out that this depends on interneuron subtype and brain region. There are a number of interneurons that mediate feedforward but not feedback inhibition, for example in the hippocampus Schaffer collateral associated interneurons and neurogliaform cells both receive extrinsic excitatory afferents but not axon collaterals of local excitatory cells. In addition, for a “hypothetical average” interneuron in hippocampus CA1, it is estimated only ~10-20% of the total excitatory input is recurrent. (See table 26, Bezaire and Soltesz 2013, https://www.ncbi.nlm.nih.gov/pmc/articles/PMC3775914/).
> > >
> > > Thank you for raising these points, we agree it would be beneficial to include a discussion of model and anatomy connectivity and propose to do so in the camera ready version of the paper if accepted.

---

### Official Review · AnonReviewer2 · 2020-11-02
**Showing that Dale's principle does not hurt the performance of feedforward ANNs does not illuminate its computational purpose**

**Rating:** 6
**Confidence:** 3

**Review:**

Inspired by the observations of feedforward inhibition in the brain, the authors propose a novel ANN architecture that respects Dale’s rule (DANN). They provide two improvements for training DANNs: better initialization and update scaling for synaptic weights. As a result, they empirically demonstrate that DANNs perform no worse than the ANNs that do not respect Dale’s rule.

Although, I find the contribution interesting, my enthusiasm is tempered by the following two issues:

1.	Although feedforward inhibition has its place in the brain, most connections of inhibitory interneurons with excitatory neurons are reciprocal, resulting in feedback inhibition. Therefore, feedforward inhibition seems like a secondary factor here.

2.	The DANNs are shown to be just no worse than ANNs that do not respect Dale’s rule. If biology “invested the effort” to evolve inhibitory interneurons respecting Dale’s rule, this is probably because they confer a computational advantage, not just lack of disadavantage.

The formulation of Dale’s rule on page 1 is not consistent with the current biological knowledge. A better version would be: “A neuron releases the same fast neurotransmitter at each of its pre-synaptic terminals”. Note that this does not mean that the action of a neuron is always excitatory or always inhibitory on all of its post-synaptic partners. It is possible, as often the case in invertebrates, that different post-synaptic partners have different receptors resulting in de- or hyper-polarization in different post-synaptic neurons.

Although the paper is generally well written, the authors could make it clearer. In particular, it would help if they defined symbols such as the circled dot or variables such as y when they are first used.

---

> ### Author Response · Authors · 2020-11-18
> **Incorporating recurrence and the question of computational advantage 1/2**
>
> We thank the reviewer for their comments, and are happy that they found our paper interesting. The reviewer’s comments raise important questions. Our responses to these points are as follows:
>
> *1) Although feedforward inhibition has its place in the brain, most connections of inhibitory interneurons with excitatory neurons are reciprocal, resulting in feedback inhibition. Therefore, feedforward inhibition seems like a secondary factor here.*
>
> The reviewer is correct that reciprocal/feedback inhibition is an important component of inhibition in the brain. We are not sure that it is fair to say that feedforward inhibition is a secondary factor, as there is ample evidence showing that feedforward inhibition is a critical, and plastic, regulator of responses in numerous circuits across the brain (see e.g. Pouille et al. 2009, Nature Neuroscience, 12:1577, 2009 or Hennequin et al. 2017, Annual Review of Neuroscience, 40:557-579 for a review related to plasticity). Indeed, the evidence suggests that feedforward inhibition can be the critical factor for determining early responses in neural circuits (Pouille & Scanziani, 2001, Science, 293: 1159–1163). Of course, learning of feedback inhibition is also important, particularly for maintaining dynamic balance and for shaping responses over time (as also explained in Hennequin et al. 2017). Thus, the reviewer is correct that including feedback inhibition would be ideal. Importantly, though, we note though that our formulation for DANNs can still be applied to feedback inhibition. Recurrent neural networks obey many of the same mathematical principles as multi-layer feedforward neural networks (see e.g. Liao and Poggio, 2016, https://arxiv.org/abs/1604.03640). If we imagine unrolling a recurrent neural network with separate excitatory and inhibitory populations, then the feedback inhibition could be treated exactly like feedforward inhibition, but with “layers” corresponding to timesteps. Thus, all of our mathematical formulations and analyses would still hold for the unrolled recurrent network. Given this important point, if the reviewer agrees, we will add a section in a revised version of the Appendix explaining how our formulation of DANNs can be used to model feedback inhibition.
>
> *2) The DANNs are shown to be just no worse than ANNs that do not respect Dale’s rule. If biology “invested the effort” to evolve inhibitory interneurons respecting Dale’s rule, this is probably because they confer a computational advantage, not just lack of disadavantage.*
>
> This is a very interesting issue that the reviewer raised, and it generated a lot of discussion amongst the authors. After discussing the matter, what we would say is that it is unclear whether Dale’s principle represents an “investment of effort” by biology or not. Though it is easy to think about possible ways to avoid Dale’s principle using known physiological mechanisms, it may also represent an evolutionary local minima, whereby early phylogenetic choices led to constraints on the system that were difficult to evolve away. This is the opinion of some of the authors. However, the reviewer may also be right that Dale’s principle does confer a computational advantage to real brains, which is why evolution kept it around. This is, in fact, the opinion of the majority of the authors. We think that future work should investigate potential advantages to Dale’s principle more thoroughly. However, this was not the goal of this study, which was instead to solve the problem of ANNs with separate excitatory and inhibitory units performing worse when trained with gradient descent. Indeed, it is hard to see how we can understand the potential computational advantages of ANNs that obey Dale’s principle if they are actually poor at learning relative to normal ANNs. Thus, we see our work as a necessary first step to future studies that could more thoroughly explore potential advantages to Dale’s principle. If the reviewer thinks it is important to include in the paper, we would add discussion of this matter to a revised version.
>
> *3) The formulation of Dale’s rule on page 1 is not consistent with the current biological knowledge. A better version would be: “A neuron releases the same fast neurotransmitter at each of its pre-synaptic terminals”. Note that this does not mean that the action of a neuron is always excitatory or always inhibitory on all of its post-synaptic partners. It is possible, as often the case in invertebrates, that different post-synaptic partners have different receptors resulting in de- or hyper-polarization in different post-synaptic neurons.*
>
> We agree with the reviewer, thank you for noting this point. We will adjust the language in the introduction to recognize the fact that there are neural circuits where the same neurotransmitters can affect different postsynaptic neurons differently.

---

> > ### Author Response · Authors · 2020-11-18
> > **Incorporating recurrence and the question of computational advantage 2/2**
> >
> > *4) Although the paper is generally well written, the authors could make it clearer. In particular, it would help if they defined symbols such as the circled dot or variables such as y when they are first used.*
> >
> > Yes, we agree. We will update the text to ensure all symbols are properly defined.

---

### Public Comment · ~Sven_Behnke1 · 2020-11-11
**ANNs with separate excitatory and inhibitory units are not new**

ANNs with separate excitatory and inhibitory units are not new.
One example of these is the Neural Abstraction Pyramid, which observed Dale's principle by using separate specific excitatory and fewer unspecific inhibitory units.

The Neural Abstraction Pyramid is a hierarchical recurrent convolutional neural architecture, for which unsupervised learning of multi-level feature hierarchies and supervised training for multiple computer vision problems,
including object detection, semantic segmentation, and image reconstruction has been demonstrated.

Key idea of the Neural Abstraction Pyramid is to iteratively incorporate partial interpretations as context in order to resolve local ambiguities.

The best reference to the Neural Abstraction Pyramid is Sven Behnke: "Hierarchical Neural Networks for Image Interpretation",
LNCS 2766 , Springer, 2003:
https://www.ais.uni-bonn.de/books/LNCS2766.pdf

---

> ### Author Response · Authors · 2020-11-11
> **The issue is training...**
>
> Thanks for letting us know about your work. Looks interesting! We will look over it to see how it applies to our work.
>
> However, please note, we never claim in the paper that ANNs with separate inhibitory and excitatory units are novel. We know others have done that before.  The key point for our paper is that ANNs with separate E and I neurons typically don't learn *as well as* standard ANNs when you apply gradient descent to them. Our paper addresses this by developing techniques for getting good gradient-based learning even when there are separate E and I populations.
>
> This is our key contribution, not simply having separate E and I populations.

---

### Author Response · Authors · 2020-11-18
**A wonderfully constructive set of critical reviews**

We thank all four reviewers for their thoughtful and insightful critiques/comments. They not only got us to examine the specific capabilities of our models more closely, they also initiated a number of interesting discussions between the authors.

We are in the process of re-writing the manuscript in order to incorporate all of the reviewer’s points. Below, the reviewers will find our responses to their specific comments which highlight the changes that we propose to make. They can also find preliminary extended results referenced in the comments here: (https://pdfhost.io/v/AFBEMscCX_DANN_Preliminary_Responsepdf.pdf ). Depending on the reviewer’s opinions of these proposed changes, we will upload a new version of the paper by Monday the 23rd, with these changes incorporated. We believe that with these modifications the paper will be stronger, and we thank the reviewers for their time and help on this.

---

### Author Response · Authors · 2020-11-23
**Paper updated**

Again, we wish to thank the reviewers for their insightful comments on our paper. We have made the modifications we proposed and uploaded a new version of the paper. The new text is highlighted in purple in the revised pdf. We note that we have included the preliminary results for the extension to convolutional networks in the Appendix, but we expect to have more thorough results ready for a camera-ready version of our paper, if it is accepted. We feel that the paper is greatly improved, and we hope that the reviewers agree.

---

### Decision · Program_Chairs · 2021-01-07
**Final Decision**

**Decision:**

Accept (Poster)

**Comment:**

This paper was unanimously rated above the acceptance threshold by the
reviewers.  While all reviewers agree it is worth accepting, they
differed in their enthusiasm.  Most reviewers agree that  major
limitations of the paper include that the paper provides no insight into why
Dale's principle exists and the actual results are not truly
state-of-the-art.  Nevertheless there is agreement that the paper
presents results worth publicizing to the ICLR audience.  The comparison
of the inhibitory network to normalization schemes is interesting.
Also, please reference the Neural Abstraction Pyramid work.